# Arctic ice clouds over northern Sweden: microphysical properties studied with the Balloon-borne Ice Cloud particle Imager B-ICI

Veronika Wolf[1], Thomas Kuhn[1], Mathias Milz[1], Peter Voelger[2], Martina Krämer[3], and Christian Rolf[3]

[1]Luleå University of Technology, Division of Space Technology, Kiruna, Sweden
[2]Swedish Institute of Space Physics (IRF), Solar Terrestrial and Atmospheric Research Programme, Kiruna, Sweden
[3]Research Centre Jülich, Institute for Energy and Climate Research 7: Stratosphere (IEK-7), Jülich, Germany

*Correspondence to:* Veronika Wolf (Veronika.Wolf@ltu.se)

August 23, 2018

**Abstract.** Ice particle and cloud properties such as particle size, particle shape and number concentration influence the net radiation effect of cirrus clouds. Measurements of these features are of great interest for the improvement of weather and climate models, especially for the Arctic region. In this study, balloon-borne in-situ measurements of Arctic cirrus clouds have been analysed for the first time with respect to their origin. Eight cirrus cloud measurements were carried out in Kiruna (68° N), Sweden, using the Balloon-borne Ice Cloud particle Imager, B-ICI. Ice particle diameters between 10 µm and 1200 µm were found and the shape could be recognised from 20 µm upwards. Great variability in particle size and shape was observed. This cannot simply be explained by local environmental conditions. However, if sorted by cirrus origin, wind, and weather conditions, the observed differences can be assessed. Number concentrations between 3 / L and 400 / L were measured, but only for two cases the number concentration reached values above 100 / L. These two cirrus clouds were of in-situ origin and were caused by gravity and mountain lee-waves. For all other measurements, the maximum ice particle concentration was below 50 / L and for one in-situ origin cirrus case only 3 / L. In the case of in-situ origin clouds, the particles were all smaller than 350 µm diameter. The number size distribution for liquid origin clouds was much broader with particle sizes between 10 µm and 1200 µm. Furthermore, it is striking that in the case of in-situ origin clouds almost all particles were compact (61 %) or irregular (25 %) when examining the particle shape. In liquid origin clouds, on the other hand, most particles were irregular (48 %), rosettes (25 %) or columnar (14 %). There were hardly any plates in cirrus regardless of their origin. It is also noticeable that in the case of liquid origin clouds the rosettes and columnar particles were almost all hollow.

*Copyright statement.* TEXT

## 1 Introduction

Cirrus clouds have a great influence on the radiation balance of the Earth and thus also on the climate (Liou, 1986; Sassen and Comstock, 2001). However, despite decades of research there are still questions which are not fully answered (Potter and Cess, 2004; Boucher et al., 2013). Open questions are for example: How are the ice particles distributed vertically? How many small

particles ($< 50\,\mu m$) are contained in a cloud and contribute to the IWC and optical properties? What are the optical properties of complex ice particle shapes? Imprecise knowledge of ice particle and cloud properties, such as particle size, shape and number concentration, leads to a remaining uncertainty about the radiation effect of the clouds and the resulting interaction with the climate. Depending on various particle and cloud properties, cirrus clouds can have a warming or also a cooling effect (Freeman and Liou, 1979; Liou, 1986; Platt, 1989; Kienast-Sjögren et al., 2016).

Particle shape and size distribution information are important for a more precise parameterisation in models to better calculate the radiant fluxes, as described by Schlimme et al. (2005). A result of their study was that particle shape has a greater influence on the optical properties of the cloud than size distribution. In addition to shape and size distribution, also roughness and hollowness of the particles are of interest, as they also influence the optical properties, as described for example by Tang et al. (2017). Gu et al. (2011) confirmed that accurate knowledge of particle properties leads to better and more realistic parameterisations and can thus improve the retrievals for remote sensing methods as well as weather and climate models.

Several studies (e.g., Lynch, 2002; Spichtinger et al., 2005; Krämer et al., 2016; Heymsfield et al., 2016, and references therein) have shown how ice cloud properties depend on meteorological and ambient conditions, such as frontal systems, waves, temperature, and humidity. Spichtinger et al. (2005), for example, described that uplift by waves not only led to an increase in supersaturation, but also to the formation of a cirrus that became optically thick within two hours. Also, Krämer et al. (2016) found different cirrus types, which are dependent on the formation mechanism and can be thicker (more IWC) or thinner (less IWC) due to the speed of the updraft. They have found that cirrus with high (low) IWC is associated with a high (low) particle concentration.

In addition to considering the local environmental conditions, ice clouds may be classified and analysed in respect to conditions at their origin (Krämer et al., 2016; Wernli et al., 2016). Certain characteristic properties may then be attributed to one of two origin types, liquid origin or in-situ origin.

The Fourth Assessment Report of the Intergovernmental Panel on Climate Change (Solomon et al., 2007) points out that improved knowledge about clouds in the Arctic is a priority because the high latitudes are much more affected by climate change than other latitudes. Due to the remoteness of large parts of the Arctic region, clouds there have been studied far less often than at other latitudes. Furthermore, most of the measuring campaigns in the Arctic have been aircraft measurements but not always especially dedicated to Arctic cirrus measurements. Campaigns in which cirrus clouds were investigated are, for example, POLSTAR 1997 (Schiller et al., 1999), FIRE-ACE 1998 (Lawson et al., 2001), INTACC 1999 (Field et al., 2001), ASTAR 2004 (Gayet et al., 2007), M-PACE 2004 (Verlinde et al., 2007) and ISDAC 2008 (McFarquhar et al., 2011).

Ice particle sampling by aircraft suffered from shattering effects at the instrument inlet due to high aircraft speed (e.g., Korolev et al., 2011, 2013; Jackson et al., 2014). This shattering led to incorrect size distributions with too many small particles. A new inlet design and algorithm might overcome this problem, at least partly (Korolev et al., 2013; Jackson et al., 2014). Another problem with aircraft measurements where the instrument is fixed under the wing is that the air around the wing is compressed and in order to calculate the number concentration, the temperature and pressure must be corrected to match the ambient conditions (undisturbed) (Weigel et al., 2016). Balloon-borne measurements avoid both these issues. An additional advantage

of balloon-borne measurements is that vertical cloud profiles can be measured with high spatial resolution. Furthermore, it is possible to measure with very high image resolution.

This study discusses balloon-borne measurements of particle properties with a particular emphasis on particle shape and size. For this, particles were imaged with a very high image resolution (1 pixel = 1.65 µm) so that the shape is identifiable from a size of 20 µm upwards. For aircraft measurements, in comparison, the often used optical array probes record the shadow of particles with pixel resolutions between 10 µm and 25 µm (Knollenberg, 1981; Lawson et al., 2006; Baumgardner et al., 2017). The cloud particle imager, CPI (Lawson et al., 2001) has at 2.3 µm a comparable pixel resolution so that it may be used for smaller particles, if they are in-focus.

The balloon-borne in-situ measurements have been carried out north of the Arctic Circle in Kiruna, in order to obtain high-resolution images of ice particles in cirrus clouds and thus provide accurate information on Arctic cirrus clouds, their particles, and properties. The measured cirrus clouds have been sorted according to their cloud origin, the meteorological situation, and the wind direction. The analysis focuses on ice particle shape, size and number concentration in relation to these conditions. The following sections describe the measurements and the instruments used. The collected data are then presented, analysed and discussed. Finally, the results are summarized.

## 2 Campaign description

### 2.1 Location and general meteorological conditions

Balloon-borne in-situ cirrus measurements have been carried out at Esrange Space Centre (ESRANGE), which is a rocket range and research centre 40 km east of Kiruna. Kiruna (68° N, 20° E) has a subarctic climate as it is located north of the Arctic Circle and east of the Scandinavian Mountains.

During winter months the conditions are influenced by the Arctic polar vortex, which is highly variable on the northern hemisphere. Kiruna is often close to the edge or inside the polar vortex with low temperatures in the lower and middle stratosphere. However, the weather as well as the polar vortex are also influenced by the positions of the planetary Rossby-waves that determine the mid and high-latitude weather. In early winter, but even later, the weather situation is usually still very unstable with a stronger influence of the low pressure systems along the polar front, leading to wind mostly from westerly directions, along the mountain range, but even from the south-east pushing air masses from the Baltic sea over the north of Sweden. Under stable conditions, that usually occur later in winter, with Kiruna being close to or inside the polar vortex, winds from westerly directions prevail and lead the air masses over the Scandinavian mountain range. Over Kiruna and ESRANGE the increased chance for orographically induced gravity waves and mountain lee waves lead to observations of related cloud formations. Additionally, under stable winter conditions, e.g. due to the influence of the Arctic and Siberian high pressure, the low amount of sun-light leads to a continuous radiative cooling that causes low temperatures in the lower and middle troposphere. This leads to very strong ground inversions, and approaching frontal systems often dissolve. All measurement days are in the winter season between the end of November and the beginning of April. Above ESRANGE during this time of the year, the minimum

temperature in the troposphere during the measurement days was between $-70°$ C and $-55°$ C. Meteorological conditions on these days are described in Sect. 3.1.

## 2.2 Measurement methods

For balloon-borne measurements of cloud and ice particle properties, an in-situ imager, the *Balloon-borne Ice Cloud particle Imager (B-ICI)*, and a radiosonde have been utilised. For a typical measurement, both are carried by the same balloon, ascending at an average vertical speed of approximately 4 m/s, through the troposphere and up to an altitude of about 13 km. The balloon type used is a plastic balloon (Raven Aerostar 19000 ft$^3$). Auxiliary data from two LIDARs, one located at Swedish Institute of Space Physics (IRF, Kiruna) and one at ESRANGE, as well as a RADAR located at ESRANGE are also used. The heart of these measurements is the in-situ imager. This device with related methods and the instruments to support the measurements are described in this section.

### 2.2.1 In-situ imager

The in-situ imager B-ICI was built for this campaign and is a light-weight (approx. 3 kg) probe for balloon-borne use. In an experiment, while ascending through the vertical extent of encountered ice clouds, it captures ice cloud particles and images them optically with a high resolution CCD camera. The images are stored on a memory card for post-flight analysis. Then, at a height of about 13 km, the instrument is cut off from the balloon and descends with a parachute back to ground, where it can be recovered. All measurements reported here were carried out during winter months when ground was covered by snow and lakes were frozen. This allowed safe landings and subsequent easy recovery of the instrument payload and image data by helicopter.

The balloon-borne probe B-ICI has been described by Kuhn et al. (2013) and Kuhn and Heymsfield (2016), however, for clarity details of the instrument will be provided here, too. Figure 1 shows the top view of the instrument with removed covers. It consists of two main units: the ice particle collecting and imaging unit which comprises the inlet (label 'a' in Fig. 1), oil-coated film (b), and part of the imaging optics (microscope objective, mirror, and illuminating LED); and the control unit comprising battery, camera (c), motor (d) and computer (e). As the imager is ascending under the balloon, ice particles enter through the inlet. The inlet opening is approximately 31 mm × 31 mm, so that at any moment a 31 mm long section of the oil-coated film is exposed to cloud ice particles. The 4 m long film is continuously moving at constant speed (1.1 mm/s) to expose always new, un-used film and avoid superposition of particles. The film is 8 mm wide and centred under the inlet, so that air will pass around the film on either side. Directly beneath the inlet, an opening on the lower side of the instrument's collecting unit with the same dimensions as the inlet allows air to move through the collecting unit. Ice particles entering directly above the film, due to their inertia, do not follow this air stream around the film and collide with it instead. Thus, these ice particles are collected, and due to the oil-coating will stay on the film. The collection efficiency has been discussed by Kuhn and Heymsfield (2016) and is 50 % at approximately 12 μm and 80 % at around 25 μm and higher for larger particles.

A camera system images the film 38 mm from the inlet. Hence, ice particles on the film are photographed shortly after collection. This camera system consists of a microscope objective, a tube lens, and a CCD sensor (1280 × 960 pixels). The

imaging optics has a high pixel resolution of 1.65 µm/pixel and an optical resolution of approximately 4 µm (as judged from the smallest details that can be discerned on the images).

### 2.2.2 Image processing

After recovery of the instrument and its image data, images are retrieved from the memory card for the following image processing on an office computer. In the first step of the three-step image processing procedure, particles are traced manually aided by a graphical computer program. This step could not be automated yet due to effects of the oil coating creating both shadows and bright regions around ice particles. In the second step, these outlines are filled and images are converted to binary masks with *true* pixels representing ice particles (belonging to one of the filled outlines) and *false* pixels representing background pixels not belonging to any filled outline. Ice particles on the binary masks are identified and their edges are found (with the Matlab function bwboundaries). Then, particle size, area, area ratio, and number concentration are determined from these particle edges. This second step is carried out automatically.

As a measure of particle size, we use a particle maximum dimension, Dmax. Several different definitions of particle maximum dimension are in use in the literature (see for example Wu and McFarquhar, 2016, and references therein) and we have chosen the diameter of the smallest circle that encloses the whole particle. The number concentration $N$ is determined from the number of ice particles collected on a given area of the film. The conversion accounts for the instrument's sample flow rate of approximately 130 cm$^3$/s and has been described by Kuhn and Heymsfield (2016). Particle size distributions d$N$/dDmax are then derived from $N$ in size bins (equally spaced on linear size scale) by dividing $N$ by the size bin width. It should be noted here, that the sampling flow rate of B-ICI, and with that also the sample volume, is independent of the particle size for sizes above approximately 25 µm, where sampling efficiency approaches 100 % (see above). For most aircraft-mounted probes such as the optical array probes this is not the case and sample volume directly depends on particle size, in particular for particles below about 200 µm in size, which results in large uncertainties in the sample volume depending on the choice of particle maximum dimension (Wu and McFarquhar, 2016). Our choice of maximum dimension is the one recommended by Wu and McFarquhar (2016), and, due to the size-independent sample volume here, it does not have an important impact on the derived number concentrations and size distributions.

In the third step, ice particles are classified manually into shapes by looking at each individual ice particle. The high-resolution images of ice particles allow us to identify shapes of particles with sizes of approximately 20 µm (12 pixels) or larger. Each ice particle is assigned to one of five shape groups: *compact*, *irregular*, *rosettes*, *plates*, and *columnar* particles. These groups were defined based on a classification by Bailey and Hallett (2009). Figure 2 shows cases of each group. Compact particles have no pronounced features deviating from a compact geometry and include particles of spheroidal shape. Rosettes include all types of bullet rosettes, column rosettes, sheath rosettes and irregular rosettes. Rosettes can have two or more arms. Plates and columnar particles are symmetrical with simple hexagonal geometries. They will most likely attach to the oil-coated film with one of their facets having the longest dimension. Thus, we classify particles visible with a hexagonal basal facet as plates and ice particles that show the longer prism facets as columnar. In addition to hexagonal columns, the shape group of

columnar particles also includes single bullets. Irregulars are those particles that cannot be sorted into any other group. For each measured cirrus cloud, all particles were assigned to one of the five shape groups.

The vertical resolution of particle number concentrations and size distributions depend on the number (or number concentration) of collected particles. In case of high particle number concentration, averaging over 10 s is sufficient, which corresponds to around 40 m vertically. For the size distributions, a slightly higher averaging period has to be used. If the particle number concentration is low, only one size distribution over the whole cloud can be averaged. This results in different vertical resolutions for the different measurement flights.

The sizing accuracy can be estimated by assuming an effective error of a few pixels when tracing the outline of ice particles. For small particles with about 20 μm (12 pixels) in size this error may be estimated as 2 pixels corresponding to approximately 17 % sizing error. For larger ice particles the error can be on the order of 3 pixels or 5 μm, which corresponds to 10 % for a 50 μm ice particle and 5 % and less for 100 μm or larger ice particles. This is similar to the experimentally determined sizing error of 4 % by Kuhn et al. (2012), who used comparable imaging optics.

### 2.2.3 Radiosonde, LIDARs and RADAR

A radiosonde is connected to the in-situ imager B-ICI. It measures temperature, humidity, altitude and geographical position. Thus, these parameters can be assigned to the photographed ice particles. The RS92 from Väisälä is used for these measurements. If available, the data from parallel observations by a RADAR and two LIDARs located in Kiruna and surrounding are also used. ESRAD, an atmospheric Mesosphere-Stratosphere-Troposphere RADAR (Kirkwood et al., 2007) located at ES-RANGE provides information on the dynamic state of the atmosphere, winds, and waves. Whenever LIDAR measurements are possible, these are used to complement the in-situ balloon-borne measurements. One LIDAR is located at the Swedish Institute of Space Physics (IRF) (about 30 km away from ESRANGE) and another one is located at ESRANGE close to the balloon launch pad. The LIDAR at IRF (Voelger and Nikulin, 2005) is an elastic backscatter LIDAR and at ESRANGE a Raman-Mie LIDAR (Blum and Fricke, 2005). The backscattered signal is used in this study as complementary information to assess the temporal and spatial characteristics of the ice clouds sampled with B-ICI. The extinction coefficients retrieved from LIDAR measurements compare favourably with the extinction measurements of B-ICI (Kuhn et al., 2017). The LIDAR beam and B-ICI sample the cloud at two locations close to each other. However, a certain distance remains resulting in an uncertainty when comparing extinction coefficients directly. An additional uncertainty arises from the fact that the LIDAR ratio (extinction coefficient/ backscatter coefficient) is not known. The in-situ data may help to constrain the LIDAR ratio, which will be tested in future with more joint data from our ongoing campaign.

### 3 Classification of Measurements

Data from eight measurement flights are presented in this study. The following subsections describe the classification of the clouds probed on these days. In Tab. 1 the flight times of the balloons and the classification of the cirrus clouds by weather

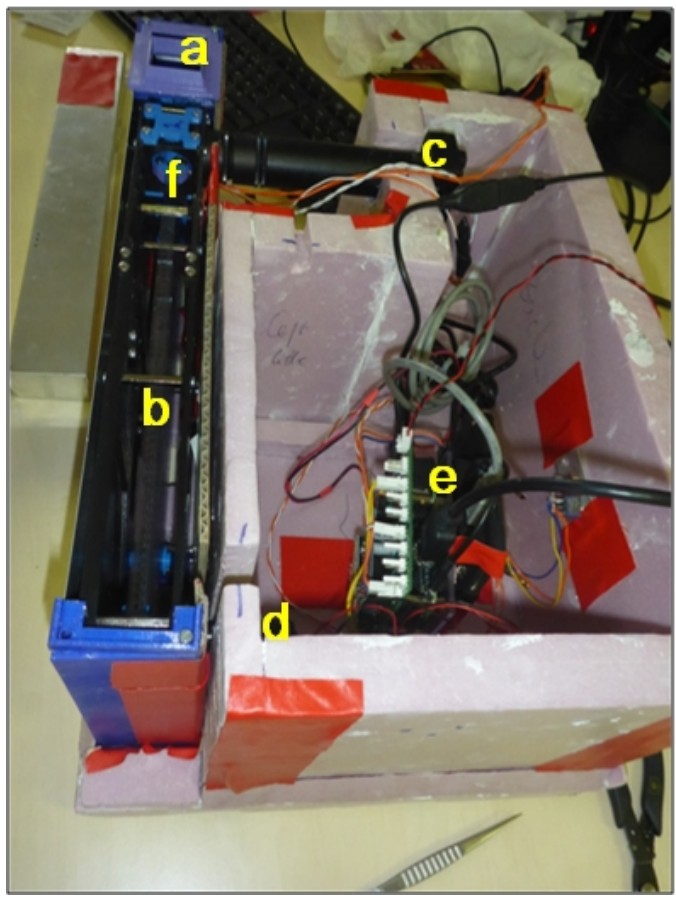

**Figure 1.** Top view on the open in-situ imager B-ICI. a) inlet where particles enter during measurement. b) oil coated moving film. c) CCD camera with an objective takes grey-map pictures every 1s. d) Motor which moves the film. e) Computer and battery f) LED to illuminate the film.

conditions (see Section 3.1) and formation origin ( see Sect. 3.2) are listed. In Sect. 3.3, the microphysical properties of the observed ice clouds are presented.

### 3.1 Weather conditions

Weather conditions on the measurement days have been analysed using weather maps, such as ground pressure with frontal analysis (from DWD) and 500 hPa geopotential (accessed on www.wetter3.de), and IR satellite images (from MSG-Eumetsat accessed on http://www.woksat.info/wos.html). The wind direction is ascertained with the help of the balloon trajectories and the back trajectories of the air mass. Figure 3 shows the back trajectories of air parcels at middle cloud heights (arithmetic mean between the bottom and top of the cloud) for 24 h before the flight (left) and the trajectories of the in-situ imager flights (right). It can be seen that the wind came from the south only on two of eight days. On all other days, the wind direction was

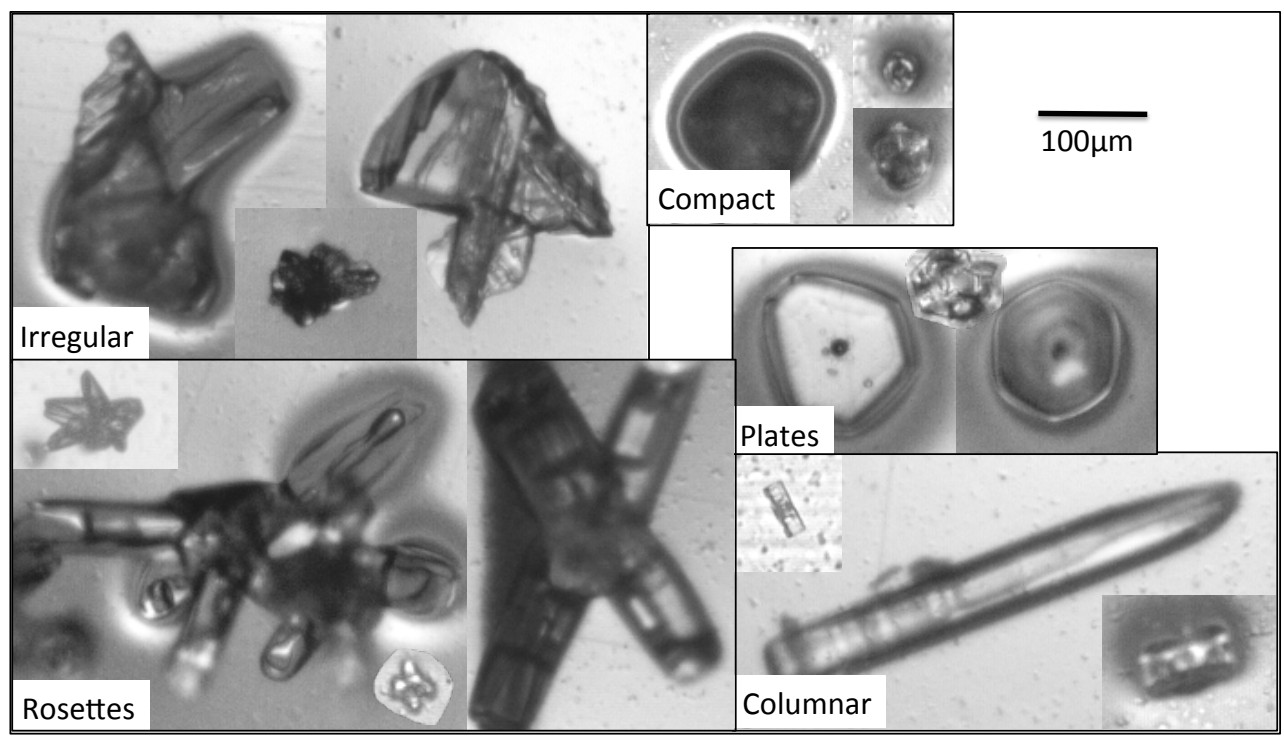

**Figure 2.** Classification of the different particles into five shape groups, which are: compact, irregular, columnar, plates, and rosettes.

**Table 1.** List of measurement days, launch and cut-off times, cloud height, mean temperature, cloud origins and meteorological situations

| date | flight time start - cut off UTC | cloud height base - top m | T °C | origin | meteorological situation | wind direction | waves |
|---|---|---|---|---|---|---|---|
| 2012-04-04 | 12:09 - 13:08 | 5550 - 7270 | -49.5 | in-situ | occlusion | NW | waves |
| 2013-02-20 | 11:15 - 12:17 | 8980 - 10440 | -62.3 | in-situ | orographic/ before cold front | NW | waves |
| 2016-03-15 | 08:26 - 09:38 | 8950 - 11550 | -56.5 | in-situ | orographic/ before cold front | NW | waves |
| 2016-12-15 | 10:03 - 11:04 | 10120 - 11750 | -65.5 | in-situ | occlusion | NW | no waves |
| 2013-12-18 | 10:45 - 11:46 | 7960 - 8050 | -52.6 | liquid | occlusion | NW | waves |
| 2014-03-20 | 12:39 - 13:42 | 6020 - 8630 | -49.3 | liquid | warm front | NW | no waves |
| 2015-04-01 | 09:40 - 10:34 | 1940 - 8410 | -34.2 | liquid | low pressure centre/ after occlusion | SSW | no waves |
| 2016-02-12 | 09:38 - 10:42 | 3400 - 10640 | -38.7 | liquid | warm front | SSW | no waves |

north-west and thus over the Scandinavian Mountains. In this case, mountain lee-waves or gravity waves can occur. Indications for this have been observed by ESRAD or LIDAR on four days. For one day (2013-02-20) Fig. 4 shows the LIDAR extinction coefficient (left) and the ESRAD vertical velocity (right). The RADAR can yield vertical velocities based on the Doppler shift of the backscatter signal. The variation of vertical velocities over time and altitude show very clearly that there were waves present at that time, horizontal wind direction points to the mountain range as source. In the case of LIDAR, the extinction coefficient shows the appearance and disappearance of clouds and the slope of clouds (inclination of cloud stripes on the altitude-time plot) indicates waves.

The cirrus was caused four times in context with an occlusion and twice in relation to a warm front. Twice the cirrus was formed in front of a cold front due to strong wind and orographic uplift over the Scandinavian Mountains.

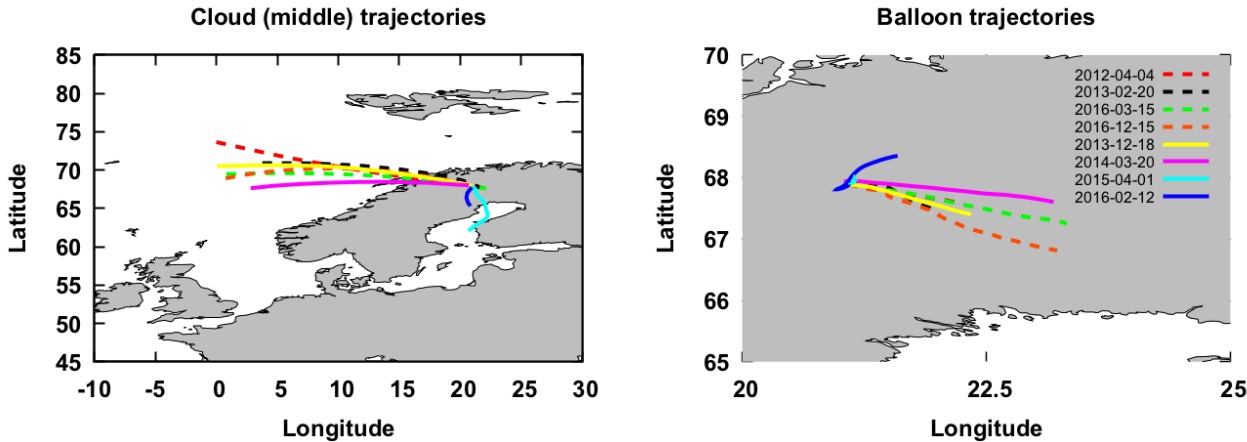

**Figure 3.** On the left the 24 h back trajectories of the average cloud height air mass for all days are shown and on the right the trajectories of the balloon measurements (dashed lines - in-situ origin, solid lines - liquid origin). The air mass back trajectories' latitudes and longitudes are calculated by CLaMS. The balloon coordinates were measured by the RS92 sonde.

## 3.2 Cirrus origin

A simple and quite new method of classifying clouds is based on their origin. Two possible cirrus origins are distinguished, *liquid* and *in-situ*. This classification is described in detail by Krämer et al. (2016) and Luebke et al. (2016) and is briefly outlined in the following. If the cloud was formed at a temperature below 235 K, it is assumed to be an in-situ origin cloud, in which particles form directly from the gaseous phase to the solid phase. If the temperature at the formation of the cloud was above 235 K, it is considered to be a liquid origin cloud. In this case the ice particles formed at lower altitudes via the liquid phase and were lifted subsequently to the cirrus temperature range. As formation in this context, we consider the time when the ice water content (IWC) started to be greater than zero, or 24 h before the in-situ measurement in case IWC was greater than zero during these 24 h.

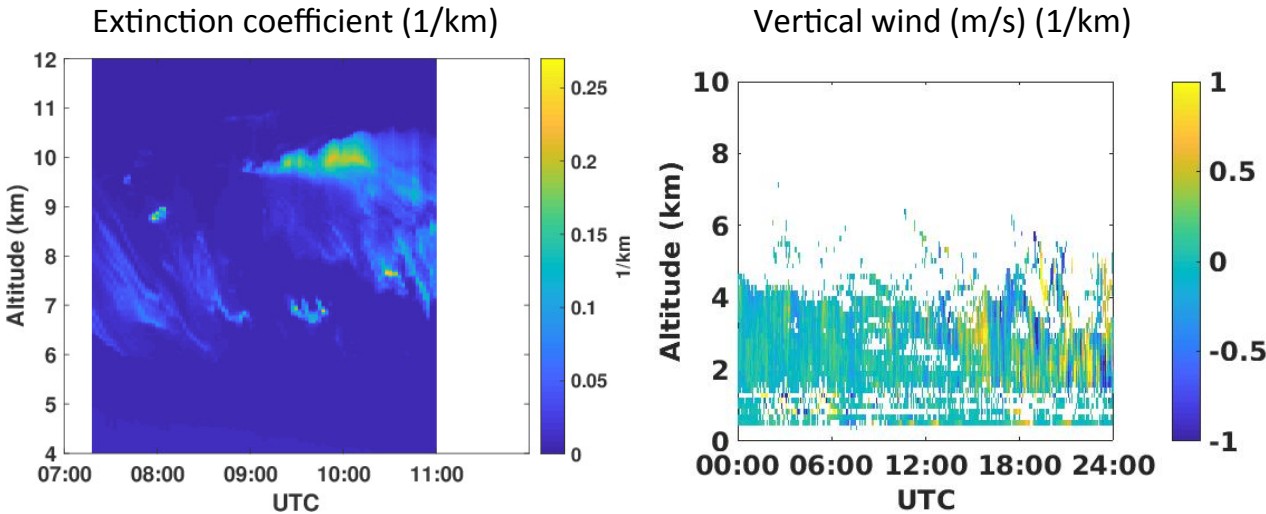

**Figure 4.** Extinction coefficient in km$^{-1}$ (left) derived from IRF LIDAR and vertical velocity in m/s (right) obtained from ESRAD on 20.2.2013.

Consequently, the cirrus origin was determined here using temperature and IWC along 24 h back trajectories. The Lagrangian microphysical model CLaMS-Ice (Luebke et al., 2016) was used to calculate these trajectories, starting from locations along the balloon flight paths, based on ECMWF ERA-Interim meteorological fields as input. Temperature was interpolated onto the trajectories, whereas the IWC along the trajectories was simulated with CLaMS-Ice. The origin of the observed cirrus cloud

was identified as in-situ if the temperature of the trajectory was always below 235 K. In case the temperature was originally higher than 235 K and carries already ice water at the time temperature crosses 235 K towards colder values, the observed cirrus is assigned as liquid origin. The resulting classifications are listed in Tab. 1. Half of the measured cirrus clouds are classified as in-situ origin, the other half as liquid origin.

### 3.3 Cloud properties

The cloud extent and averaged temperature for each measurement are listed in Tab. 1 and Fig. 5 shows the corresponding temperature and humidity profiles. The lower altitude of the first cloud level and the upper altitude of the last cloud level define the total extent of the cloud. Two cirrus clouds (2015-04-01 and 2016-02-12) had a vertical extension of approximately 6 km with a low cloud base at an altitude of 2 km and 3 km, respectively. It may not be correct to call these clouds cirrus. However, in both cases, the entire cloud contained ice phase only, and the lower levels represent, as will be discussed later,

glaciated, previously mixed-phase clouds. We believe these to be interesting cases and included them in our cirrus study. The other six cirrus clouds were thinner (80 m – 2 km thick) and had a higher cloud base (over 6 km). In all cases the temperatures decreased with altitude. The temperatures at the cloud tops were between -60° C and -70° C. At the cloud base, the temperatures were between -45° C and -55° C in case of thin clouds and between -10° C and -20° C in case of the two thick clouds. The

relative humidity with respect to ice in the clouds was between 80% and 130%. Particles with sizes between 10 μm and 1200 μm were collected. Smaller particles are not efficiently sampled (Kuhn and Heymsfield, 2016), and larger particles have not been encountered. Table 2 lists the size ranges and mean number concentration for each cloud. The ice particle number concentrations were between (3/L) and (400/L) and the profiles of the number concentration for each measurement day are shown in Fig. 8. For each measured cirrus cloud, the frequency of occurrence of shapes is summarized in Tab. 2 as percentages corresponding to the five particle shape groups compact, irregular, rosettes, plates and columnar. Some images of the particles from each measurement are shown in Fig. 6. All particle images are shown with the same size scaling.

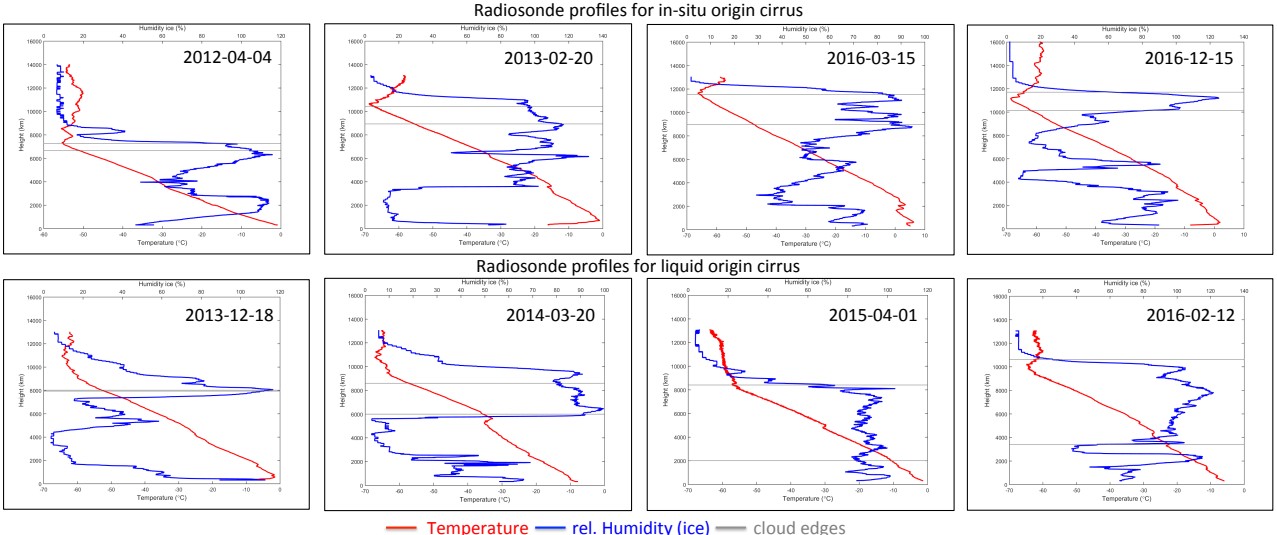

**Figure 5.** Temperature (red) and relative humidity profiles (blue) with respect to ice for the eight measurement days (upper row in-situ origin, lower row liquid origin). The clouds upper and lower edge are marked by horizontal lines.

## 4 Results and Discussion

### 4.1 Size and Number concentration

On two days (2015-04-01 and 2016-02-12, see Tab. 2 and Fig. 7) we collected very large ice particles, with maximum sizes of approximately 600 μm and 1200 μm respectively. Both days represent two liquid origin cases with southerly winds, low cloud base (totally frozen, previously mixed-phase cloud), and large vertical extension. In cases of in-situ origin cirrus, all particles were smaller than 350 μm. On three of the four days with in-situ origin (2013-02-20, 2016-03-15 and 2016-12-15) all particles were even smaller than 100 μm. This difference in size is also reflected in the number size distribution (PSD). Figure 7 shows PSDs for all measurement cases, where possible also for different height levels. The in-situ origin PSDs are fairly narrow, which indicates that the corresponding clouds are quite young, homogeneously formed cirrus, where the ice crystals have not

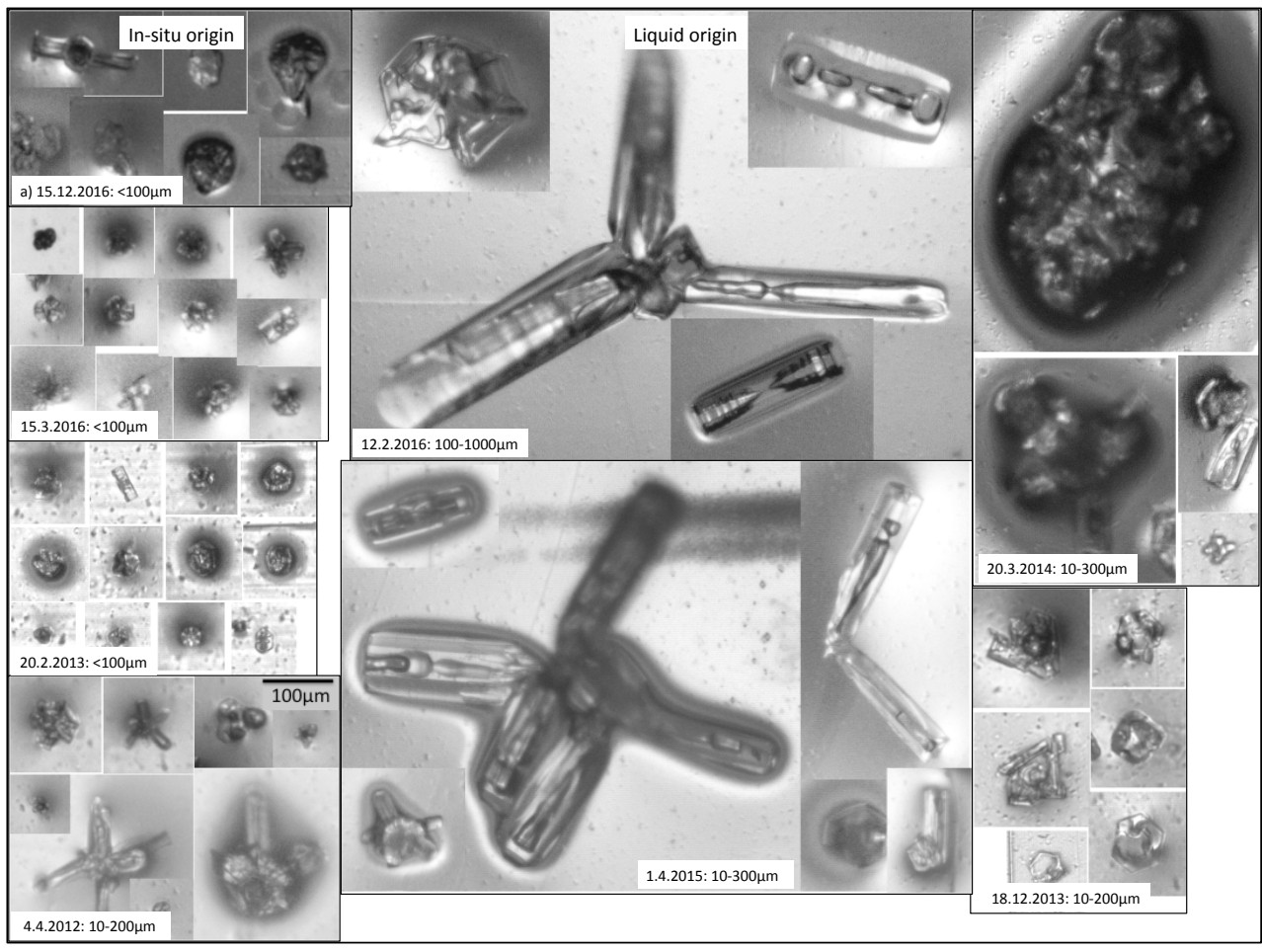

**Figure 6.** Some pictures of ice particles from all measurement days. The left panel shows ice particles from in-situ origin cirrus, on the right liquid origin crystals are displayed. For a better understanding of the size, a 100 µm bar is displayed (2012-04-04 bottom). All images have the same scale resolution and 100 µm corresponds to 61 pixel.

**Table 2.** List of days, mean number concentration $N$, mean particle Dmax, and relative number (in %) of particles in different shape groups.

| Date - origin | $N$ | Dmax min/ median/ max | compact | irregulars | rosettes | plates | columnar |
|---|---|---|---|---|---|---|---|
| | 1/L | µm | % | % | % | % | % |
| 2012-04-04 - in-situ | 37 | 7/ 96/ 327 | 38.8 | 42.2 | 16.4 | 0.4 | 2.2 |
| 2013-02-20 - in-situ | 228 | 14/ 34/ 91 | 71.7 | 19.3 | 2.5 | 1.8 | 4.7 |
| 2016-03-15 - in-situ | 13 | 25/ 41/ 105 | 72.4 | 21.3 | 4.6 | 0.7 | 1.0 |
| 2016-12-15 - in-situ | 3 | 25/ 52/ 102 | 63.9 | 16.8 | 4.2 | 0.0 | 15.1 |
| 2013-12-18 - liquid | 16 | 24/ 84/ 277 | 4.3 | 69.6 | 6.5 | 8.7 | 10.9 |
| 2014-03-20 - liquid | 38 | 11/ 100/ 492 | 27.2 | 60.8 | 9.5 | 0.3 | 2.2 |
| 2015-04-01 - liquid | 8 | 22/ 201/ 643 | 7.2 | 23.9 | 49.3 | 1.6 | 18.0 |
| 2016-02-12 - liquid | 6 | 5/ 244/ 1228 | 6.0 | 39.2 | 28.5 | 3.7 | 22.6 |

yet grown to larger sizes. On 2012-04-04, the ice particles may have grown somewhat more than on the other days with in-situ origin clouds, leading to somewhat wider PSDs on that day.

All distributions of the liquid origin clouds extend to larger sizes and are broader than in the case of the in-situ origin. In order for the particles to grow that large, a sufficiently high temperature with the related high water vapour concentration is required. Such conditions are given for liquid origin clouds. The PSDs of the two liquid origin clouds originating from the south (2015-04-01 and 2016-02-12) are particularly wide. The other two liquid origin clouds have almost similarly narrow PSDs as the in-situ origin clouds and were probably already in the process of dissolving, and large particles lost via precipitation. On 2013-12-18 for example, the cloud was very thin (80 m) and the relative humidity (ice) above and below the cloud was even strongly under-saturated. Many of the collected particles looked as if parts had already sublimated.

In general, the PSDs are more narrow and the number concentration (NC) higher with increasing height and decreasing temperature. This can be clearly seen, for example, for the in-situ origin clouds in Fig. 7 and Fig. 8. The dependence of in-situ origin PSDs and NC on altitude and temperature is likely due to the main ice nucleation zone being at the cloud top. This dependence of the PSDs and NC has also been found on a global scale (e.g. Sourdeval et al., 2018; Gryspeerdt et al., 2018). This PSD and NC trend with altitude and temperature is not clearly seen for the liquid origin cirrus cases. This could be explained by the fact that liquid origin cirrus form at lower altitudes and then ascent in the prevailing updraft. The few ice particles, nucleated in warmer and thus also moister air masses, grow to large sizes which sediment out of the air mass while ascending. This means that in pure liquid origin cirrus there is no process enhancing the number concentration of smaller ice particles towards the cloud top or higher altitude.

However, these variations in PSDs with altitude or temperature are less than the general differences observed between in-situ origin and liquid origin. The broadest size distribution at the lowest height of the in-situ origin cloud on 2013-02-20 for example is still much more narrow than any distribution of the liquid origin cloud on 2016-02-12. This is true also for the other measurement days, even when considering the two liquid origin clouds with more narrow PSDs, which are still broader than

in-situ PSDs at similar temperatures. That means that size distributions measured in different clouds but at similar altitudes and temperatures can be significantly different. While these differences are obviously not only related to the local ambient conditions, they are strongly related to the cloud origin.

Data reported earlier from aircraft measurement at high latitudes also show a large range in sizes comparable to the observations of our balloon measurements. Gayet et al. (2007) described a measurement in which they collected falling ice particles from a cirrus cloud above. The size distribution between 25 μm and 1000 μm mentioned by them corresponds well with our two thick liquid origin clouds, which had their lower edge approximately at the same height and similar temperature. Furthermore, Sourdeval et al. (2018) presented PSDs from five aircraft campaigns (ATTREX, ACRIDICON-CHUVA- tropics, SPARTICUS, ML-CIRRUS and COALESC - mid-latitudes). In the averaged data, they observed, in addition to a primary mode of sub 100 μm particles that was always present, a secondary mode of larger than 100 μm particles that appeared only at temperatures higher than -50°C. They discuss that this large particle mode is due to liquid origin cirrus. Thus, a comparison with our measurements shows that in the Arctic liquid origin clouds with larger particles can still occur at lower temperatures.

So far, only Krämer et al. (2016) and Luebke et al. (2016) investigated the dependence of size on cloud origin for mid-latitude spring cirrus. For Arctic cirrus, our observations corroborate their findings that in-situ origin clouds contain smaller particles than liquid origin clouds. Furthermore, one can recognize in the number size distributions in Fig. 7 that in the case of in-situ origin cirrus clouds, NC can be many times higher than for liquid origin. The total number concentrations are shown in Fig. 8 as altitude profiles for all eight measurement days (top: in-situ origin, bottom: liquid origin). Comparing 2013-02-20 and 2016-02-12, it can be seen that, the maximum concentration of the in-situ origin cloud was approximately 20 times greater than in the liquid origin cloud. This much higher number concentration in the case of this in-situ origin cloud does not apply to all of our in-situ origin cases, but only to two measurements (2013-02-20 and 2012-04-04). On 2016-03-15 the NC was similar as in case of liquid origin measurements and on 2016-12-15 the NC was very low with just 3 / L.

Such high differences in NC between in-situ origin clouds may be related to the influence of wave activity. Krämer et al. (2016) discussed two types of in-situ origin cirrus. The first type appears in slow updrafts, e. g. in warm conveyor belts. The ice is nucleated mostly heterogeneously and the corresponding ice particle number concentrations are low. In the second type which is related to fast updrafts, the ice particles form homogeneously with high number concentrations triggered by the fast updraft. The two days (2013-02-20 and 2012-04-04) with higher number concentration (300 -400 / L) and also the 2016-03-15 were associated to very strong wind coming from the north-west which led to waves, as observed by ESRAD or LIDAR on both days. These gravity or mountain lee waves with the related high vertical velocities can be the needed trigger for such high number concentrations (e.g. Lohmann and Kärcher, 2002). Field et al. (2001) showed that number concentrations in wave clouds can even rise with decreasing temperature up to 100000 / L. In contrast, the mid-latitude in-situ origin cirrus observations, described by Krämer et al. (2016) and Luebke et al. (2016), showed lower NC than those in the Arctic. The reason may be that most of the Arctic in-situ observations are influenced by mountain waves with high vertical velocities triggering homogeneous nucleation of many ice crystals. Such observations were very rare in Krämer et al. (2016) and Luebke et al. (2016).

The number concentrations of the liquid origin clouds were always relatively low (5 / L to 70 / L). The lower NC in comparison to Luebke et al. (2016), who found a median ice number concentration slightly above 100 / L in liquid origin mid-latitude cirrus, might be due to a lower number of ice nucleating particles (INP) in Arctic regions (Costa et al., 2017), which are necessary for heterogeneous freezing. However, low number concentrations could also be caused by a dissolving cloud state. To confirm this, one would need INP or humidity measurements during some time before our measurements, hence, we can only speculate here.

In the discussion above, we have noticed that values for NC were different compared to values reported for cirrus in the mid-latitudes. While in liquid origin clouds the NC was lower, it was higher in in-situ origin clouds compared to the same cloud type in the mid-latitudes. To understand this and the general differences between the two origin types better one can look at the different formation pathways and thus differences in microphysical properties of these two cirrus origin types. The in-situ cirrus clouds formed at temperatures below 235 K either by hetereogeneous nucleation of ice nucleating particles (INP) or homogeneous nucleation of super-cooled solution particles (Krämer et al., 2016). Thus, the ice particle number concentration of such clouds is in the range of the available INP or given by the large number of homogeneously nucleated particles in this temperature range due to fast updrafts.

Number concentrations are on average smaller than in liquid origin clouds. However, in fast updrafts, many small ice crystals form homogeneously, which is reflected in a high number concentration and was more often the case in our measurements. As a result, in our case three of the in-situ origin clouds had higher or about the same NC as liquid origin clouds. Liquid origin clouds are present typically in case of convection or large scale transport like warm conveyor belts. The cirrus ice particles of liquid origin are mostly formed by heterogeneous freezing at lower altitudes and temperatures above 235 K, where typically mixed-phase clouds occur. They are uplifted into the in-situ temperature range where they at latest fully glaciate. In the original altitude, more water vapour and INPs are available, resulting together with the continuous updraft in larger particles, higher number concentration and thus higher ice water content compared to the in-situ origin clouds with slow updraft. As indicated earlier, differences between mid- and high latitudes may then be explained by differences in the available INP for liquid origin clouds and the larger influence of waves on our measurement cases in the Arctic in case of in-situ origin.

## 4.2 Shape

In our individual cases there is no significant dependence of the shape on temperature and relative humidity (with respect to ice). Furthermore, no particular dependence of particle shape over the height was found. Therefore, we are reporting the average frequency of occurrence of the different particle shapes (see Tab. 2) and discuss how that varies depending on cloud origin. These average frequencies of shape occurrence for in-situ origin and for liquid origin clouds are also shown in Fig. 9 (left panel). The right panel of this figure shows how the average particle sizes of the different shapes vary depending on the cloud origin. As can be seen in Fig. 6 and Fig. 9, in the case of in-situ origin, the particles are usually small in size and compact or irregular in shape. However, in the case of liquid origin, the particles are most commonly irregular and rosettes. In-situ origin clouds form at a temperature range (< -38°C) where the water concentration in the atmosphere is very low. Therefore, there is

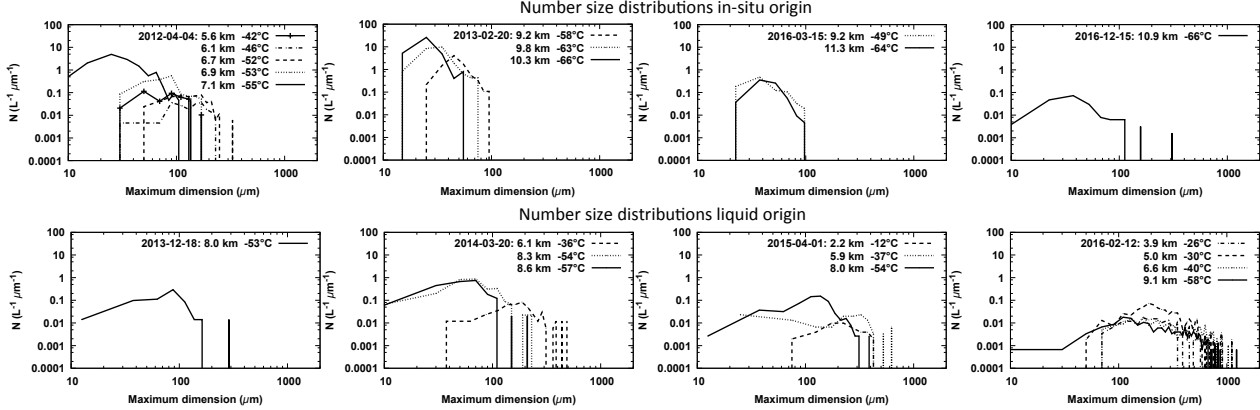

**Figure 7.** Number size distributions for all measurement days for different cloud levels (in-situ origin top and liquid origin bottom).

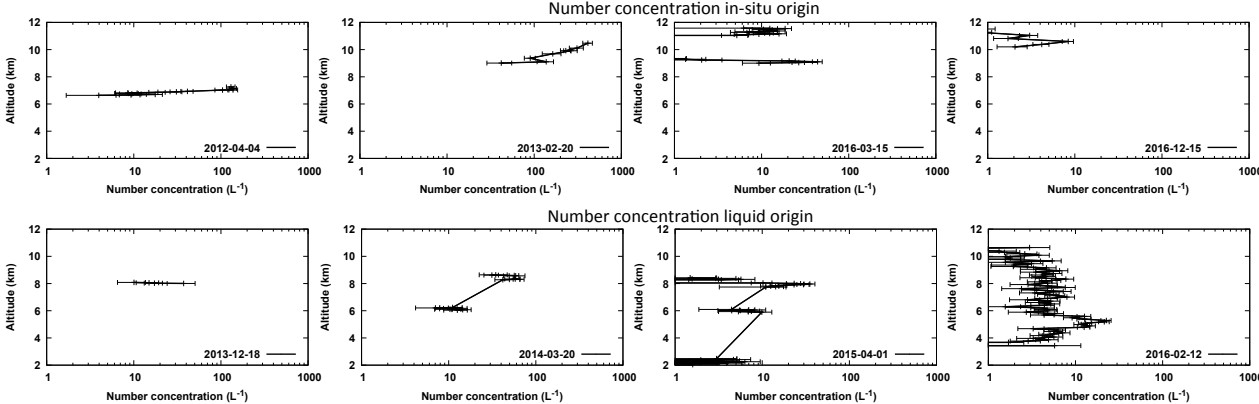

**Figure 8.** Number concentration as a function of altitude for all measurement days (in-situ origin top and liquid origin bottom)

not enough water available to form large or complex shapes. Hence, it is understandable that most of the in-situ origin cirrus particles found were compact.

While compact particles are on average the smallest ones, rosettes, irregular and columnar particles in liquid origin clouds were largest. Liquid origin cirrus clouds, in contrast to in-situ origin cirrus, form at warmer temperatures with higher water vapour content in the air. Therefore, the ice particles can grow larger and also to more complex shapes. Particularly large ice particles were observed on the two days (2015-04-01 and 2016-02-12) where the lower part of the cloud was in the temperature regime of mixed-phase clouds. As discussed earlier, at the time of measurement these two clouds were completely frozen. However, the liquid water, which was probably present at some earlier stage, has contributed to the observed extensive growth. This is in agreement with Bailey and Hallett (2009), who claim that a high supersaturation is needed for the growth of rosettes and hollow columns, which were abundant on those days. Fewer rosettes were found on the other two days (2013-12-18 and

2014-03-20). This may be unexpected, however, it may be explained by larger particles falling out of the probably ageing clouds. In fact, these clouds looked like they were in the process of dissolving, as discussed earlier in Sect. 4.1.

It is noticeable that almost all columnar particles and rosettes were hollow in case of liquid origin cirrus. This corroborates findings by others (e.g. Weickmann et al., 1948; Heymsfield et al., 2002; Schmitt et al., 2006), in which measurements showed that around 80 % of all collected rosettes were hollow to a certain extent. In the case of in-situ origin cirrus there are very few, and if present, then very small rosettes and columns. Thus, a statement regarding their hollowness would be rather speculative.

The supersaturation present in our liquid origin cloud measurements is most of the time too low to directly explain growth of our observed hollow rosettes and columns. According to laboratory measurements by Bailey and Hallett (2004), existence of hollow rosettes requires high supersaturation, and hollowness of rosettes is more likely at higher temperatures (> -40°C). While the temperature and water vapour at which the particles were detected is too low, ambient properties at the origin of the clouds met the conditions for hollow rosette growth in the case of liquid origin clouds. Thus, this demonstrates once more that environmental conditions at cloud origin are crucial for explaining observations.

In both origin cases, plates and columnar particles were rarely collected. They are on average less frequent than any of the other shapes. This is similar to Korolev et al. (1999), who have collected only 3 % of these shapes. In Tab. 2 it can be seen that in the case of in-situ origin clouds on 2013-02-20 the highest percentage of plates was sampled with only 1.8 %. Somewhat more columns were collected, on average 5.7 %. In the case of liquid origin these particle shapes were on average more frequent than in the case of in-situ origin, as can be seen in Fig. 9. Plates were on average 3.0 % of all observed ice particles, and columns are with 12.3 % even a little more frequent than compact particles (10.8 %).

Shape detection is sometimes intricate, even with high image resolution. Some particle shapes may be confusing, as also observed by others (e.g., Lindqvist et al., 2012). Here, the assignment between irregulars and rosettes was sometimes ambiguous, because in a few cases rosettes appear somewhat irregular. For example, some rosettes look as if they have a part missing or one bullet seems to be a longer column. In such cases, we have assigned these irregular rosettes to the shape group rosettes rather than to irregulars. In other cases, small compact ice particles sometimes show characteristics that indicate an initial formation of rosettes, however, we have still classified them as compact due to their spheroidal shape. Classifying them as rosettes would not have changed any of the results discussed here.

For ice particles smaller than 20 µm the shape is difficult to recognize and, consequently, some misclassification may occur leading to over-representation of compact in this size range and under-representation of other shapes such as plates and rosettes. However, on the day with the smallest particles (2013-02-20) only about 6 % of all particles were smaller than 20 µm. Thus, this issue of potential misclassification will likely not alter our findings significantly.

## 5  Summary and Conclusions

In this study, eight balloon-borne in-situ measurements of Arctic cirrus clouds were analysed. The balloons were launched from Kiruna, Sweden during winter time. Particular emphasis was placed on the analysis of ice particle size, shape and number concentration with respect to cirrus origin. Since in-situ origin clouds are formed from the gas phase at temperatures below

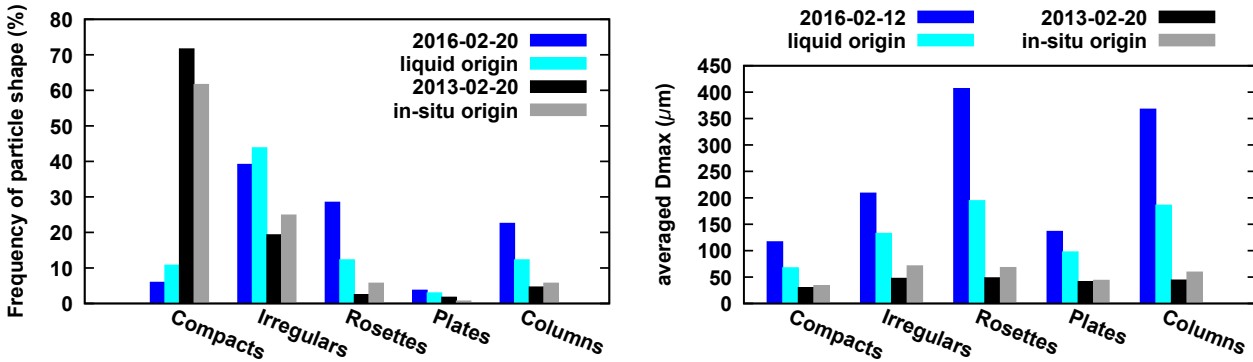

**Figure 9.** Occurrence of different particle shapes depending on cloud origin (left) and average Dmax for the different shapes (right). Mean values of shape and size of the four in-situ origin (gray) measurements and four liquid origin (light blue) measurements. Mean values of shape and size of one in-situ origin measurement day (black) and of one liquid origin measurement day (blue).

235 K, while liquid origin clouds formed via liquid drops at temperatures above 235 K, the cloud and particle properties are expected to vary in accordance to cloud origin. And indeed, while large differences in particle size, shape and number concentration are observed between the various measurements, some similarities are noticed within the two groups of data with liquid and in-situ origin clouds, respectively. These similarities and the differences between data, when grouped in liquid and in-situ origin, are summarized below:

1. Particle size: Arctic cirrus clouds with particle sizes between 10 μm and 1200 μm have been observed. Most common in our clouds are particles with sizes between 30 μm and 250 μm. While in-situ origin clouds have smaller particles with sizes below 350 μm, liquid origin clouds exhibit larger particles and wider number size distributions. The ice particles of clouds with wind from the south are much larger and fewer than ice particles from the west where the cirrus was probably triggered by strong updrafts associated with gravity or mountain lee waves behind the Scandinavian Mountains.

2. Particle shape: The in-situ origin clouds consisted mainly of compact and irregular particles and the liquid origin clouds of irregular, rosettes and columns. In both cases, there are hardly any plates. The compact particles were the smallest particles and rosettes were the largest. Rosettes and columns were mostly hollow.

3. Particle number: The measured number concentrations were between 3 / L and 400 / L. Both extreme values were determined for in-situ origin clouds. The higher concentrations occurred due to waves on the lee side of the Scandinavian Mountains. In comparison, in previous campaigns in the mid-latitudes lower number concentrations were measured for this cloud type. This may be explained by the fact that hardly any wave-induced in-situ origin clouds were observed in these campaigns. Concentrations for liquid origin clouds were low (5 / L to 70 / L). In contrast, high number concentrations were measured in the mid-latitudes for this cloud type, maybe caused by a higher number of INPs in the mid-latitudes than in the Arctic.

The results of this study imply that remote sensing retrievals and weather and climate models could be improved when accounting for these differences rather than using parameterisations that depend only on local conditions. Future work will include more measurements for further significant statistical evaluation. In addition, we also want to allow several B-ICIs to fly one after the other in order to investigate a temporal development of the particle properties.

5  *Acknowledgements.*  We thank Peter Dalin (IRF) and Evgenia Belova (IRF) for interpreting and discussing the ESRAD data in terms of gravity and mountain lee-waves. We thank the Swedish National Space Agency for funding these balloon campaigns (Grants Dnr 85/10, Dnr 86/11, Dnr 143/12, Dnr 273/12, Dnr 168/13, Dnr 124/14).

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
