# Peer review of "Arctic ice clouds over northern Sweden: microphysical properties studied with the Balloon-borne Ice Cloud particle Imager B-ICI"

_Atmospheric Chemistry and Physics, 2018_

## Short Comment (SC1) · 4 Jun 2018

Manuscript: Ice particle properties of Arctic cirrus Referee comments: Overall: This manuscript needs to be improved significantly. There are many issues related to text flow and scientific understanding of the Arctic cirrus clouds, check on cirrus dynamics from SHEBA project. . ... Results are also contradictory for the theory of parameterizations and needs to be clarified. More cases wrt satellite and lidar/radar should be used and connected to IC concentrations. Presently content is poorly written and not discussed based on other works in the Arctic clouds. Specifically, liquid origin and local origin concepts are misleading formation of these clouds. There are many issues with this paper and they are listed as: 1. abstract is not given explicitly; no info on what kind of balloon being used? 2. what sensors are used? 3. no meaning of liquid clouds at cirrus level? Not good naming, and very confusing. 4. in-situ origin cloud? Cirrus form due to IN and its properties are related to local or advection. 5. how do you explain the

liquid origin and local origin? This doesn't make sense; and you don't have a mechanism to explain it. 6. 61% compact??? And 25% irregular, is this a resolution issue? Seems to me it is resolution issue unless you have a proof of it. 7. page2; no shattering at this level because already they are small, take out refs on this. Balloon is not like airplane. .8. what parameterizations? 9. "we detect particles. . .." no you don't, sensor does. 10. depends on ambient conditions. . . . . . do not include waves, systems, and temperature together. . .. Confusing and not meaningful. What is role of T wrt waves or systems. Talk about its physics, T ok. 11. For these reasons????? What reasons? 12. introduction is confusing and not clear. 13. location; what level (height) measurements were taken? Is this cirrus or arctic BL cloud? 14; what is the in-situ imager? Imager of what? name should be ice crystal imaging probe or similar. . ... ICIP???? Check your earlier works, it says differently. 15. what is the compact means? I feel these are not resolved particles, out of focus particles. 16. page 4; lidar extinction? You should include some work here on this. 17. radar and lidar images were not clearly used to support cirrus dynamics. But they should. Not enough to say water origin or local origin. Table 1 should state height levels. Figure 2; size of these particles should be in the image. Again, what is the meaning of compact? Page 6; shows how did you use satellite images, show a case. Page 7; smaller particles are not efficiently sampled. . ... how small? Page 8; Table 2; at >-60C, you have more IN, why you have these??? But not always true? It is against IN parameterizations, explain it. Fig 4; liquid origin? How do you know? Page 10; higher than this in liquid origin? Why? This is against the nature of formation again. Figure 5; what is the uncertainty in Ni measurements? and what is the time period for collection of Ni? How did you calculate Ni? Figure 6; this figure useless; need to show sampling time, and number of points used in Ni calculations. Need to show all other cases. Ni is calculated what? TAS? Sampling area? Etc. Fig. 7; you need to show calculation of ext here. Also you need to show at least cases with extreme conditions such as Ni~5 and Ni~300 L-1, and then discuss it. Fig. 7b; why the Vd given at the BL is important for cirrus level? Don't you have a figure for cirrus level? You need a comparison table or figure for outcome of this work. Then explain

what the results are significantly different.

---

## Referee Comment (RC1) · Anonymous Referee #1 · 11 Jun 2018

**1. Overview of the paper:**

This paper presents balloon borne in situ measurements of cirrus clouds over the Kiruna region. Eight "flights" are analysed to derive the vertical distribution of microphysical properties (shape, size, and number concentration of ice crystals) of cirrus. Cirrus clouds are classified according to their origin: namely in situ-origin or liquid-origin. The main results show a variability in particle size, shape and to a lesser extent number concentration. This variability seems to be mainly connected to the cirrus origin. The observations presented in this study are useful and the topic is relevant. New measurements of the vertical properties of ice crystals within cirrus clouds are important, especially if they are combined with information on the dynamical state of the atmosphere. I like the idea of linking the microphysical properties to the in situ or liquid origin of cirrus. It gives researchers a framework for comparing cirrus properties in different region of the world and to understand dynamical process responsible for the

formation of cirrus clouds. The balloon-borne observations of the vertical distribution of cirrus microphysical properties are potentially very useful for the community. However, a more thorough data analysis and a better presentation of the results should be done before considering the publication of the paper in ACP. I would recommend major revisions.

Below I have compiled a list of general comments and more specific comments that should be considered (hopefully) in a revised version of the paper. Not all are mandatory but I have the feeling that at least some could help to improve the readability of the manuscript.

2. Major comments:

Data analysis and interpretation

I have the feeling that the authors could do a better job in the analysis of their measurements. The results are not always presented in a clear and coherent way. Sometimes, the data analysis does not fully support the conclusions drawn by the authors. All the measurements should be presented and compared (figure 5 and figure 6). Most of the main findings are based on only 2 or 3 cases. A more thorough interpretation of RADAR and LIDAR observations should be done to support the conclusions. The main conclusions on the impact of cirrus origin on microphysical properties should be detailed. The authors jump to conclusions without discussing (or showing) the entire dataset. I also would expect a small discussion including comparison with previous findings at mid latitude and in the Arctic. The authors should also explain what is their definition of a cirrus clouds since ice layers at -20C/2000m are considered.

General structure of the paper :

The text is sometimes not easy to read. I would suggest that the authors seek for an additional proof reading. As I am not a native English speaker (as you can see), I will not go into details to point out grammar errors as I might be mistaken. The general

structure of the paper could be modified to improve the manuscript clarity. Some figures would need a more thorough discussion and interpretation. I would reorganise section 3 and section 4 to focus on the results of the study. Then, a section called "discussion" should be added where the results could be compared to previous findings at mid latitude and in the Arctic. Lidar and Radar measurements should be presented in this section and a more complete analysis should be performed. Finally, the last section should be called summary and conclusions.

3. Specific comments :

0. Title

"Ice particle properties of Arctic cirrus" might not be the most appropriate title for this study. I would recommend the authors to be more specific as the case studies presented in the paper are not proven to be representative of all cirrus found in the Arctic. An alternative title could be "Vertical microphysical properties of Arctic cirrus over the Kiruna region (68°N, X°E)".

1. Introduction

The introduction could be significantly improved to deliver a clearer message. Editing and reorganisation of sentences and paragraphs would be appreciated. Some statements/sentences should be clarified and completed.

Page 1 - Lines 21-22: I think that you should state the main questions to be answered here. For instance: What are the sedimentation velocities and the optical properties as a function of the ice crystal shape and complexity? What is the relationship between IN and ice crystal concentration? How is the vertical distribution of size and shape in cirrus clouds? What is the contribution of small ice crystal (D<50$\mu$m) to the IWC? What is the spatial scale of cirrus properties inhomogeneities? Etc...

Page 2 – Lines 3-5: Are you sure that IPCC points out that the improved knowledge of cirrus clouds properties in the arctic is a priority. I think that low level clouds such as

mixed phase clouds are also a large (larger?) source of uncertainties in models. You might want to slightly change that sentence.

Page 2 – Lines 6-8: There has been a lot of airborne campaigns carried out in the Arctic focusing on clouds or aerosol-cloud interactions. Recently, ACCACIA-2013, ACLOUD-2017 were performed in the European Arctic region. POLARCAT 2008, ASTAR 2004 & 2007, SORPIC 2010 also took place over the Norwegian Sea- Greenland Sea region. Other campaigns were also undertaken in the Western Arctic region such as: ISDAC-2008, M-PACE 2004, FIRE-ACE 1998, ARCPAC 2008, VERDI 2012, RACEPAC 2014 . . .. Some of these campaigns should be cited in the introduction. They might not have focused on cirrus clouds but I'm pretty sure that some measurements of cirrus cloud properties were performed

Page 2 – Lines 9-10: please rephrase and shattering should be introduced later in your introduction (see comments below).

Page 2 – Lines 12-18: This paragraph is important as it presents some of main results from modelling activities as well as some of the key properties to assess. It should be moved to line 5-p2 or page 1.

Page 2 – Lines 25-30: This paragraph should be positioned before the paragraph on airborne measurements. Moreover, it would be good if you could briefly summarize the main results obtained by Lynch et al..... Kramer et al. . . ..

Page 2 – Line 34: Could you be more specific when you write "the analysis focuses on ice particle and cloud properties" ? What do you mean? ice crystal shape and size ?

2. Campaign description

2.1 Location

Page 3 – Lines 5-9: At this point, I would recommend giving more details on the meteorological conditions (synoptic and maybe local), to discuss the influence of the Scandinavian mountains on cloud formation and properties and to describe more precisely the

measurement period (indeed, measurement days are mentioned but are not indicated at this point).

**2.2 Measurement methods**

Page 3 - Line 11: "for the measurements of cloud and particle properties" what do you mean here by particle properties? I did not see any aerosol measurements in the paper? Or do you mean cloud particle properties? You should also specify that the in situ imager is balloon-borne. Some details should also be given on the type of balloon.

**2.2.1 In situ imager**

Could you give more details on the sampling method, efficiency, shortcomings and potential measurement errors linked to the instrument and the fact that it is balloon-borne?

Does the in situ imager has a name? Maybe you should replace in situ imager by cloud particle imaging probe. What is the weight of the instrument?

Page 3 Lines 23-24 : I think you should use the past tense in this sentence (was / were instead of is/are). What do you mean by partly manually partly automatically? Could you be more specific and elaborate on the reasons why this cannot be done with a fully automatic algorithm (are you talking about the ice crystal shape classification or pre processing of the data to check for acceptable non distorted images etc, see also my comment on figure 2 ) ?

Page 3 Line 25 : What do you mean by "Once the particle outlines have been traced"?. You should also explain briefly how the microphysical parameter were calculated from your images and with which accuracy.

Page 3 Line 27 : "smallest diameter of the circle that encloses the whole particle" is this the diameter of the smallest circle that encloses the ice crystal? Could you give some references on how this maximum dimension compares to other diameters used in Optical Array Probes ?

Page 4 Line 1 : Compact particle are spheroidal : ok but you might want to use spheroidal in the abstract to avoid any misunderstanding.

2.2.2 Radiosonde, LIDARs and RADAR-LIDAR

I have the feeling that LIDAR and RADAR data could be more thoroughly exploited to complement the cirrus in situ measurements (in a discussion section for instance). As mentioned by the authors, those measurements can be used to describe the dynamical properties of the atmosphere. These additional measurements experiments would strengthen the main findings of this paper. In the present form of the paper, I don't really see the added value of such measurements (the lidar figure is not described and the radar figure needs a better description/analysis : see comment section 4 and figure 7)

Page 4 Lines 15-16 : "Radiosonde data, temperature, humidity, height and geographical coordinates can be assigned to each particle" : this sentence does not sound right. The use of the word "particle" is ambiguous. Do you mean cloud layer with a 60m vertical resolution?

Page 4 Line 24 : You should shortly sum up the main results of the in situ imager – Lidar extinction coefficient comparison. Otherwise, I don't understand the meaning of this sentence.

3. Classification of measurements

3.1 Cirrus origin

Table 1 Page 5 and Line 11 Page 6 : Table 1 is interesting but I think average Temperature and Altitude values could also be mentioned here. Could you also explain in the text which kind of weather maps and satellite images were used to describe the meteorological situation?

Figure 2 Page 6 : You mention latter in the text that the assignment between irregulars and rosettes was sometimes ambiguous. What about plate and compact spheroidal

ice crystals?. Looking at figure 2, I can imagine that it is quite hard to discriminate small compact crystals from small plates. It looks like the shadow of the coating is distorted/modified by the impact of the ice crystal on the coating. It might result in an increase of the degree of "roundness" of the ice crystal, meaning that if an automatic classification algorithm is used small ice plates could be classified as compact ice crystal (explaining that you find almost no plates in your cirrus cases). Am I wrong? Could you discuss mis classification issues? You should also show the size of the ice crystals on figure 2.

Page 6 Lines 4-5 : I think a verb is missing in this sentence, please consider rewriting this sentence.

Page 6 Lines 6-10 : You might want to clarify this paragraph. I know that you are not supposed to fully describe the methodology described in Kramer et al., 2016 and Luebke et al., 2016. However, I think it is still necessary to elaborate on this cirrus classification as it is linked to the in situ microphysical properties.

3.2 Weather conditions

Page 6 Line 13: What are the average cloud heights ?

Page 7 Line 3 : I see that now the RADAR ESRAD is mentioned and used to detect the occurrence of Lee waves or gravity waves. For my personal understanding, could you explain me how this is done?

3.3 Cloud properties

Table 2 : Table 2 is not easy to read and does not look very "attractive". But it is still quite important. I would recommend modifying it or maybe transforming it into a graph (if possible). If you want to keep that table, please use the same date format as the one used in table 1, use colours according to the air mass origin ( in accordance with figure 3).

Page7 Lines 7-10 : I'm getting lost here, I don't understand how a cirrus could have

a geometrical thickness of 6km and a cloud base close to 2km (and temperature of -11.5°C). Could you elaborate on the cirrus definition used in your study? These two thick clouds have a liquid origin and are associated with southerly winds. Looking at Kramer et al., ACP 2016, and Luebke et al. 2016 I can read that liquid origin cirrus are characterized by : (1) high IWC, high ice crystal concentration (NC>100 L-1), and large ice crystals (D>200$\mu$m) (2) nucleation mechanism is probably homogeneous freezing (low IN) (3) Fast updrafts (4) They appear with liquid containing clouds below

From your results presented in table 2, we can see that the ice crystal size is on average larger for liquid origin cirrus but the ice number concentration is very low (especially for the 01.04.2015 & 12.02.2016 case). How do you explain this? It doesn't not seem to agree with mid latitude results presented in Kramer et al., 2016 and Luebke et al., 2016. I'm also wondering if the low layers considered as cirrus clouds correspond to mixed phase clouds, glaciated clouds or fall streaks? How can you tell that low level cloud layers are solely composed of ice crystals : you have no cloud droplet measurements ?

Page 10 - Table 3 : Table 3 displays the distribution of ice crystal habits within each "flights". It is interesting but hard to compare. An indication of the temperature and relative humidity with respect to ice should be provided along these values. A vertical distribution of the cloud shape would also be more valuable. In your statistics you are "mixing" ice crystals measured at 2000m/-11°C with ice crystals found at 8000m/-54°C and compare it to ice crystals found at 11km/-65°C ? Is this relevant ?

In in situ cirrus, the fraction of compact ice crystals seems to be high (40% to 70%). Is this in agreement with previous results found in cirrus clouds? The fraction of plate is very low but don't you think it is due to a possible misclassification of small plates to compact ice crystals. Once again, this should be discussed in the paper.

4. Results and Discussion

4.1. Size and number concentration

Page 10 line 3 : "see observations 2" : what does it mean ? Maximum size displayed on table 2.

Page 10 line 5 : "At three of the four days" should be something like "During three of the four days"

Figure 5 – Page 12 : I think that you should show your results in log-log scale (with dN/dlogDmax vs Dmax for instance) – not mandatory as you might not see the difference (broadness of PSD) highlighted in the paper. However, I think an additional panel where the PSD measured at comparable temperature should also be shown. It would help support your main conclusions regarding the differences of PSD behaviour found for liquid origin cirrus and in situ cirrus.

Page 10 Lines 11-15 : It would be good if you could rephrase this paragraph to help the reader understand your point. "vastly" should be significantly. The fact that the PSD is narrower with increasing height and decreasing temperature is clearly evidenced on the in situ cirrus case. Size is decreasing and NC is increasing. The PSD is very narrow and almost look like monodispersed distribution, is it really representative? Is it due to sampling issues? This temperature/altitude trend is not clearly seen for the liquid origin cirrus case. Why ? Do you have microphysical process hypothesis to explain this behaviour?

Page 10 Line 16 : "While these differences are obviously not related to local ambient conditions, they are related to the cloud origin" : this statement might be a bit strong. Without showing additonnal cases, it is hard to be so positive... What about humidity measurements? I did not see any in the paper. It could be useful to better interpret your dataset.

Page 10 Lines 17-18 : Gayet al., 2007 focused on a case study where observations of ice crystals precipitation (from cirrus ?) down to a supercooled boundary layer stratocumulus were made. Measurements were performed at 1500m/-11°C. The PSD shows ice crystals with size ranging from 25$\mu$m to 1000$\mu$m with a Deff=270$\mu$m (and NC=10

l-1). I understand that in situ measurements in arctic cirrus are scarce but this study is hardly comparable to your study. At least you need to be more precise in comparing your results, do you mean that you are comparing the PSD of precipitating ice crystals (which case is this in your study ?) to Gayet et al., work ?

Page 10 Lines 21-23 Yes, I agree that the number concentration of ice crystals found in this in situ cirrus is higher than in the liquid origin cirrus. This is not in agreement with previous findings of Kramer et al. and Luebke et al.. I think that all your cases should be presented on Figure 6. It would be easier to see if the vertical profiles are linked to the in situ/liquid origin or the air mass origin. It is hard to draw conclusions based on two very specific cases.

Page10 Lines 23-24 : "It should be noted that the y axis . . .. in concentration" : you could delete this sentence.

Page 11 Lines 1-5 : Fig 6 is very important but I don't understand why only two cases are shown. If possible, the 8 flights should be plotted on this figure. You also say that two cases (half of your in situ cirrus events) of in situ origin cirrus cloud (20/02/2013 & 15/03/2016) exhibit high ice crystal number concentrations, sometimes much higher than concentration found in liquid-origin cirrus. It is true for the 20/02/2013 case but I don't think this the case for the 15/03/2016 where concentration is close to 11-14 l-1 on average (according to table 2). Some cases of liquid-origin cirrus reach 56 l-1 and the 04/04/2012 in situ origin cirrus concentration reaches 131 l-1 at 7km. So, I don't understand your comparison. Please, clarify this point as it does not make sense to me. Once again, this also shows that each profile should be plotted on this figure to facilitate the comparison and draw solid conclusions.

Page 11 – Lines 9-11 and figure 7 : It is a good idea to use lidar and radar measurements but I think that you need to go more into details. You show the vertical profile of the extinction coefficient measured from the LIDAR but I don't see the added value of such plot : nothing is said about it or compared (extinction, altitude, structure of the

cloud...). What about the lidar and measurements performed during the liquid-origin cirrus event?

Page 11 – Lines 10-11 figure 7 : Without a more detailed explanation it is hard to see/understand how wind vertical velocity measurements below 5km can explain "waves with high velocities can explain such higher number concentration". Please clarify this.

Page 11 -Lines 14-16 : This could be an explanation, indeed. From your results, one can see that the ice crystal sizes agree with Luebke et al. But not the concentrations. The reasons for such discrepancies should be discussed and your results should be compared to other measurements in cirrus clouds (at mid latitude and in the Arctic if there were any). I also have the feeling that the vertical distribution of Nc is much more variable for in situ origin cirrus than for liquid origin cirrus, why ? Don't you think it is a problem to compare cirrus properties at very different altitudes ? I think that you sometimes compare fall streaks, high and cold cirrus (-66°C-10000m), with warm low ice clouds (-11.5°C -2000m ) ?

Page 11 – Line 16 : should be "Arctic region"

4.2 Shape

Page 11 Lines 20-25 : This paragraph is more a discussion than actual results. It should be moved either to a new discussion section or to line 10 p 12. Your paragraph should start with "The frequency of occurrence of the different particle shape... line 26.

Page 12 Line 6 : "this corroborates findings by others" : which findings ? be more specific. It is important to compare your results with other measurements. For instance, I am surprised to see that rosettes are mainly found in liquid-origin cirrus, at which temperature? . My question is : Do you really think that the shape of the ice crystals is more likely to be influenced by the origin of the cirrus (meaning in situ or liquid) or the temperature and Rhi ?

[Figure]

Page 12 Lines 5-10 : please rephrase this paragraph, I don't understand what you are trying to show.

5. Conclusions

Page 13 Line 7 : "when looking at the cirrus in terms of its origin, similarities between the various properties are striking" : I don't understand what you mean here : you are saying just above that large differences in ice particle size, shape and number are observed and then that similarities are striking when looking at the origin of cirrus.... please rephrase.

Line 8-9 : I think this sentence should be placed after the summary of the most important results.

Page 14 : I would suggest to also summarize the comparison between your work and previous studies using the same cirrus classification.

Please also note the supplement to this comment:
https://www.atmos-chem-phys-discuss.net/acp-2018-386/acp-2018-386-RC1-supplement.pdf

---

## Referee Comment (RC2) · Anonymous Referee #2 · 27 Jun 2018

Review of
**Ice particle properties of Arctic cirrus**
by Veronika Wolf et al.

**General comment:**
In this study, arctic cirrus clouds are investigated, using measurements from balloon-borne instruments. The data from eight radiosonde ascents are investigated about shape, size and number concentration of ice particles. In combination with trajectory calculations, the formation pathway can be determined and the microphysical properties can be related to these pathways.

Overall, this is an interesting study using a very promising technique for the detection of ice particles on a very well suited platform; thus, this is an adequate and meaningful contribution to ACP. However, there are some issues which should be clarified before the manuscript can be accepted for publication. Therefore I recommend major revisions for the manuscript.

In the following I will explain my concerns in detail.

**Major points**

1. Definition of liquid origin and in situ formation not clear
   The study relies strongly on the recent developed classification scheme by Krämer et al. (2016), separating ice crystal formation pathways into liquid origin and in situ formed ice crystals. However, the definitions of these two types seem not to be correct from a thermodynamic point of view: liquid origin is characterised by formation at water saturation, while in situ formation occurs at conditions below water saturation. Please correct and extend the definitions in the manuscript accordingly, see also Krämer et al. (2016) or even Wernli et al. (2016).

2. Interpretation of data and scientific results
   While the measurements of the ice crystals show very high quality and seem to be quite interesting, the evaluation of the data is weak. It is not really, what the authors want to state with their results. Especially, the interpretation of the data concerning the different pathways is not clear. What is the story you want to tell? What did you expect for ice crystal shape, size and number concentrations for the different formation mechanisms? What is the result and how can this be interpreted? Is there any hint from theory to corroborate these findings (was it expected or surprising, and why?)? Invest more theory for the interpretation of the data and the presentation of the results. Finally, it would be nice to have figures of the profiles, at least in the appendix.

**Minor points:**

1. High speed measurements:
   Actually, high speed measurements have some other issues beside the problem of shattering, see e.g. the compression of air as indicated in the study by Weigel et al. (2016).

2. Classification of data partly manually/automatically:
   It is stated in the text, that the classification was carried out partly automatically. Please describe how this was done and which techniques were used.

3. Measurements with RADAR/LIDAR:
   What was the outcome of the complementary measurements of RADAR and LIDAR? Is there any additional value for the results/interpretation?

4. Listing of the different clouds in table 2:
   It is not clear to me, how the authors can count 4 clouds, because it seems that there are two adjacent layers, since the top layer of the first cloud (e.g. 5680m) is the same as the bottom layer of the next cloud. Please explain this interpretation.

---

## Author Comment (AC1) · 23 Aug 2018

**Final author comments**
Title: Ice particle properties of Arctic cirrus
Author(s): Veronika Wolf et al.
MS No.: acp-2018-386
MS Type: Research article

*This is the response to Referee Comment RC1.*
*Thank you very much for this assessment and the many constructive comments.*
*In the following we are responding point by point to your comments. Our responses are formatted in italics and updated text in manuscript stands in quotation marks.*

**Review of "Ice particle properties of Arctic cirrus" by V. Wolf et al., ACPD - 2018**

**1. Overview of the paper:**

This paper presents balloon borne in situ measurements of cirrus clouds over the Kiruna region. Eight "flights" are analysed to derive the vertical distribution of microphysical properties (shape, size, and number concentration of ice crystals) of cirrus. Cirrus clouds are classified according to their origin: namely in situ-origin or liquid-origin. The main results show a variability in particle size, shape and to a lesser extent number concentration. This variability seems to be mainly connected to the cirrus origin.

**The observations presented in this study are useful and the topic is relevant**. New measurements of the vertical properties of ice crystals within cirrus clouds are important, especially if they are combined with information on the dynamical state of the atmosphere. I like the idea of linking the microphysical properties to the in situ or liquid origin of cirrus. It gives researchers a framework for comparing cirrus properties in different region of the world and to understand dynamical process responsible for the formation of cirrus clouds. The balloon-borne observations of the vertical distribution of cirrus microphysical properties are potentially very useful for the community. However, a **more thorough data analysis and a better presentation of the results should be done before considering the publication of the paper in ACP. I would recommend major revisions**.

Below I have compiled a list of general comments and more specific comments that should be considered (hopefully) in a revised version of the paper. Not all are mandatory but I have the feeling that at least some could help to improve the readability of the manuscript.

**2. Major comments:**
**Data analysis and interpretation**
I have the feeling that the authors could do a better job in the analysis of their measurements. The results are not always presented in a clear and coherent way.
Sometimes, the data analysis does not fully support the conclusions drawn by the authors. All the measurements should be presented and compared (figure 5 and figure 6). Most of the main findings are based on only 2 or 3 cases.

*For clarity, only two example cases (one for in-situ and one for liquid origin) were always displayed, not all eight measurement cases. As suggested, all measurements are now*

*displayed and evaluated. This should make it clearer that the main findings are not only based on two example cases.*

A more thorough interpretation of RADAR and LIDAR observations should be done to support the conclusions.

*The LIDAR and RADAR observations are only used to assess the temporal and spatial properties of the ice clouds that have been sampled with the in-situ imager. A first comparison between in-situ imager and LIDAR measurements with regard to the extinction coefficient has already been described by Kuhn, 2017. A more thorough interpretation of the LIDAR observations will include an evaluation of the depolarization ratio (LIDAR) in comparison to particle shape. While this is planned in the future, it would go beyond the scope of this article, in which we would like to focus on the in-situ measured particle shapes, sizes and number concentration in relation to the cirrus origin.*

The main conclusions on the impact of cirrus origin on microphysical properties should be detailed. The authors jump to conclusions without discussing (or showing) the entire dataset. I also would expect a small discussion including comparison with previous findings at mid latitude and in the Arctic.

*All data sets are now displayed and the discussion has also been extended with comparisons of other studies.*

The authors should also explain what is their definition of a cirrus clouds since ice layers at -20C/2000m are considered.
*We not only evaluated cirrus but also mixed-phase clouds, which when observed were completely frozen and merged directly into cirrus above. Strictly speaking, it may not be correct to call the whole cloud cirrus. However, for simplicity we call even these thick clouds cirrus. We do this also because we don't want to exclude these layers because we think they are interesting and you can see the transition from a previously mixed-phase cloud to a liquid origin cirrus cloud. This is something, we think, has not yet been reported from aircraft measurements.*
*See also the answer to comment Page 7 Line 7-10.*
*Rather than a definition of cirrus we have included a kind of disclaimer making the reader aware of this and trying to motivate calling all clouds 'cirrus'.*
*(in Sect. 3.2: "…cloud base at an altitude of 2 km and 3 km, respectively. It may not be correct to call these clouds cirrus. However, in both cases, the entire cloud contained ice phase only, and the lower levels represent, as will be discussed later, glaciated, previously mixed-phase clouds. We believe these to be interesting cases and included them in our cirrus study.")*

**General structure of the paper :**
The text is sometimes not easy to read. I would suggest that the authors seek for an additional proof reading. As I am not a native English speaker (as you can see), I will not go into details to point out grammar errors as I might be mistaken. The general structure of the paper could be modified to improve the manuscript clarity. Some figures would need a more thorough discussion and interpretation. I would reorganise section 3 and section 4 to focus

on the results of the study. Then, a section called "discussion" should be added where the results could be compared to previous findings at mid latitude and in the Arctic. Lidar and Radar measurements should be presented in this section and a more complete analysis should be performed. Finally, the last section n should be called summary and conclusions.

*As a result of the specific comments below we have changed and improved the manuscript. Many figures have also been improved or added to facilitate the discussion of the whole data set. While the general structure has not changed, some sections are structured better and the discussion is more thorough. The presentation of the LIDAR and RADAR data has also been improved, however, it still only fulfils the goal to support certain aspects of our analysis of the in-situ data as explained in our related responses below.*
*The improved conclusion section has been called 'Summary and Conclusions'.*

**3 .Specific comments:**

i. **Title**

"Ice particle properties of Arctic cirrus" might not be the most appropriate title for this study. I would recommend the authors to be more specific as the case studies presented in the paper are not proven to be representative of all cirrus found in the Arctic. An alternative title could be "Vertical microphysical properties of Arctic cirrus over the Kiruna region (68°N, X°E)".

*True, the title was a little too general. The new title is: "Arctic ice clouds over northern Sweden: microphysical properties studied with the Balloon-borne Ice Cloud particle Imager B-ICI"*

**Introduction**
**The introduction could be significantly improved to deliver a clearer message. Editing and reorganisation of sentences and paragraphs would be appreciated. Some statements/sentences should be clarified and completed.**

ii. **Page 1 - Lines 21-22**: I think that you should state the main questions to be answered here. For instance: What are the sedimentation velocities and the optical properties as a function of the ice crystal shape and complexity? What is the relationship between IN and ice crystal concentration? How is the vertical distribution of size and shape in cirrus clouds? What is the contribution of small ice crystal (D<50μm) to the IWC? What is the spatial scale of cirrus properties inhomogeneities? Etc...

*Thank you for the suggestion. Some open questions have been included in the revised manuscript.*

*"Such open questions are for example: How are the ice particles distributed vertically? How many small particles (50μm) are contained in a cloud and contribute to the IWC and optical properties? What are the optical properties of complex ice particle shapes? This imprecise knowledge of ice particle and cloud properties, such as particle size, shape and number*

*concentration, leads to a remaining uncertainty about the radiation effect of the clouds and the resulting interaction with the climate."*

iii. **Page 2 – Lines 3-5:** Are you sure that IPCC points out that the improved knowledge of cirrus clouds properties in the arctic is a priority. I think that low level clouds such as mixed phase clouds are also a large (larger?) source of uncertainties in models. You might want to slightly change that sentence.

*Yes, right, mixed-phase clouds represent a big uncertainty for the models, too. The report mentions problems for mixed-phase clouds as well as cirrus clouds.*
*Since we have not only observed cirrus but also completely frozen mixed-phase clouds, we have now removed the reference to cirrus and refer instead to clouds in general.*

*"The Fourth Assessment Report of the Intergovernmental Panel on Climate Change ( Solomon 2007) points out that improved knowledge about clouds in the Arctic is a priority because the high latitudes are much more affected by climate change than other latitudes."*

iv. **Page 2 – Lines 6-8:** There has been a lot of airborne campaigns carried out in the Arctic focusing on clouds or aerosol-cloud interactions. Recently, ACCACIA-2013, ACLOUD-2017 were performed in the European Arctic region. POLARCAT 2008, ASTAR 2004 & 2007, SORPIC 2010 also took place over the Norwegian Sea-Greenland Sea region. Other campaigns were also undertaken in the Western Arctic region such as: ISDAC-2008, M-PACE 2004, FIRE-ACE 1998, ARCPAC 2008, VERDI 2012, RACEPAC 2014 …. Some of these campaigns should be cited in the introduction. They might not have focused on cirrus clouds but I'm pretty sure that some measurements of cirrus cloud properties were performed

*Yes, some of the campaigns mentioned by the referee include cirrus measurements in high latitudes, though unfortunately many of them are dedicated solely to liquid and mixed-phase clouds.*
*The article now mentions the campaigns that included cirrus measurements: POLSTAR 1997, INTACC 1999, ASTAR 2004, M-PACE 2004 and ISDAC 2008. These have also been used for comparisons in the discussion in the modified manuscript.*

 *"Campaigns in which cirrus clouds were investigated are, for example, POLSTAR 1997 (Schiller 1999), FIRE-ACE 1998 (Lawson 2001), INTACC 1999 (Field2001), ASTAR 2004 (Gayet 2007), M-PACE 2004 (Verlinde 2007) and ISDAC 2008 (McFarquhar 2011)."*

v. **Page 2 – Lines 9-10:** please rephrase and shattering should be introduced later in your introduction (see comments below).

*The sentence has been rearranged. This section about shattering has now been placed after the section mentioning Arctic campaigns. We have also added another issue with aircraft measurements related to sample volume uncertainties due to pressure changes below the wings where these instruments are mounted. The fact that balloon-borne measurements are not affected by any of these two issues has also been made clearer.*

*"Airborne particle measurement suffered from shattering effects at the instrument inlet*

*due to high aircraft speed." … "Another problem with aircraft measurements, described by Weigel 2016, is that the air around the wing under which the instrument is mounted is compressed. As a result, in order to calculate the number concentration, the temperature and pressure must be corrected to match the ambient conditions (undisturbed). Balloon-borne measurements avoid both these issues. An additional advantage of balloon- …"*

*Further change:*

*"This study discusses balloon-borne measurements…"*

*next paragraph: "The balloon-borne in-situ measurements have been carried out…"*

vi. **Page 2 – Lines 12-18:** This paragraph is important as it presents some of main results from modelling activities as well as some of the key properties to assess. It should be moved to line 5-p2 or page 1.

***Thank you for pointing that out, we followed the suggestion and moved the paragraph as recommended.***

vii. **Page 2 – Lines 25-30**: This paragraph should be positioned before the paragraph on airborne measurements. Moreover, it would be good if you could briefly summarize the main results obtained by Lynch et al..... Kramer et al. ….

***Also, this paragraph has been moved as suggested.***
***It is difficult to list the most important results of the literature mentioned, as some of them are books or book chapters providing summaries of the entire cirrus research field. However, a short section with results has been added.***
***"Several studies (e.g. Lynch 2002, Spichtinger 2005, Kraemer 2016) have shown how ice cloud properties depend on meteorological and ambient conditions, such as front systems, waves, temperature, and humidity, also see (Heymsfield 2016) and references therein. Spichtinger 2005, for example, described that uplift by waves not only led to an increase in supersaturation, but also to the formation of a cirrus that became optically thick within two hours. Also, Kraemer 2016 found different cirrus types, which are dependent on the formation mechanism and can be thicker (more IWC) or thinner (less IWC) due to the speed of the updraft. They have found that cirrus with high (low) IWC is associated with a high (low) particle concentration."***

viii. **Page 2 – Line 34**: Could you be more specific when you write "the analysis focuses on ice particle and cloud properties" ? What do you mean? ice crystal shape and size ?

***Yes, we can. We focused on particle shape, size and number concentration.***
***In manuscript changed to:***
***"The analysis focuses on ice particle shape, size and number concentration in relation to these conditions."***

**2. Campaign description**

**2.1 Location**

ix. **Page 3 – Lines 5-9**: At this point, I would recommend giving more details on the meteorological conditions (synoptic and maybe local), to discuss the influence of the Scandinavian mountains on cloud formation and properties and to describe more precisely the measurement period (indeed, measurement days are mentioned but are not indicated at this point).

*The weather situation is explained in Part 3.2 (Weather conditions) and listed in Table 2. Only the location was described here. Following your recommendation, we have changed the title of this subsection to "Location and general meteorological conditions" and a general description of the weather has now been added.*

*"Balloon-borne in-situ cirrus measurements have been carried out at Esrange Space Centre (ESRANGE), which is a rocket range and research centre 40 km east of Kiruna. Kiruna (68°N, 20°E) has a subarctic climate as it is located north of the Arctic Circle and east of the Scandinavian Mountains.*

*During winter months the conditions are influenced by the Arctic polar vortex, which is highly variable on the northern hemisphere. Kiruna is often close to the edge or inside the polar vortex with low temperatures in the lower and middle stratosphere. However, the weather as well as the polar vortex are also influenced by the positions of the planetary Rossby-waves that determine the mid and high-latitude weather.*

*In early winter, but even later, the weather situation is usually still very unstable with a stronger influence of the low pressure systems along the polar front, leading to wind directions mostly from the southwest, along the mountain range, but even from the southeast pushing air masses from the Baltic sea over the north of Sweden.*

*Under stable conditions, that usually occur later in winter, with the location being close to or inside the polar vortex, winds from westerly directions prevail and lead the air masses over the Scandinavian mountain range. Over Kiruna*
*and ESRANGE the increased chance for orographically induced gravity waves and mountain lee waves lead to observations of related cloud formations.*

*Additionally, under stable winter conditions, e.g. due to the influence of the Arctic and Siberian high pressure, the lack of sun-light leads to a continuous radiative cooling that causes to low temperatures in the lower and middle troposphere. This leads to very strong ground inversions, and approaching frontal systems often dissolve.*

*All measurement days are in the winter season between the end of November and the beginning of April. Above ESRANGE during this time of the year, the minimum temperature in the troposphere during the measurement days was between -70°C and -55°C. Meteorological conditions on these days are described in Sect. 3.2."*

**2.2 Measurement methods**

x. **Page 3 - Line 11**: "for the measurements of cloud and particle properties" what do you mean here by particle properties? I did not see any aerosol measurements in the paper? Or do you mean cloud particle properties? You should also specify that the in

situ imager is balloon-borne. Some details should also be given on the type of balloon.

*We mean properties of ice particles and were not referring to other types of particles. This has been made clearer in the manuscript. It has also been mentioned here (and been made clearer in the introduction) that the measurements are balloon-borne with the type of the balloon specified (Raven Aerostar 19000 cf plastic balloon).*

**2.2.1 In situ imager**
**Could you give more details on the sampling method, efficiency, shortcomings and potential measurement errors linked to the instrument and the fact that it is balloon-borne?**
Does the in situ imager has a name? Maybe you should replace in situ imager by cloud particle imaging probe. What is the weight of the instrument?

*The in-situ imager has now a name. It is Balloon-borne Ice Cloud particle Imager (B-ICI). The recommended name cloud particle imaging probe (- CPIP) is in our opinion too similar to the CPI.*
*The weight of approx. 3kg has now been mentioned in the manuscript. We have also included some more details about this new balloon-borne probe.*

xi. **Page 3 Lines 23-24** : I think you should use the past tense in this sentence (was / were instead of is/are). What do you mean by partly manually partly automatically? Could you be more specific and elaborate on the reasons why this cannot be done with a fully automatic algorithm (are you talking about the ice crystal shape classification or pre processing of the data to check for acceptable non distorted images etc, see also my comment on figure 2 ) ?

*We have added more details about the image processing procedure in the manuscript. Part of this description is still in present tense since it is a general description of the procedure. However, the tense has been changed in the sentence that has been pointed out.*

xii. **Page 3 Line 25** : What do you mean by "Once the particle outlines have been traced"?. You should also explain briefly how the microphysical parameter were calculated from your images and with which accuracy.

*As mentioned above, we have described the image processing procedure better in the manuscript, so this should be clearer now. Sizing accuracy is now also discussed and estimated.*

xiii. **Page 3 Line 27** : "smallest diameter of the circle that encloses the whole particle" is this the diameter of the smallest circle that encloses the ice crystal? Could you give some references on how this maximum dimension compares to other diameters used in Optical Array Probes ?

*Yes, it is the diameter of the smallest circle that encloses the ice crystal, thank you for pointing this out. We have changed the manuscript accordingly.*
*There are several definitions of maximum dimension used for Optical Array Probes (OAP). In case of the OAP, the sample volume depends strongly on the particle size, in particular below about 200 µm. This means that the choice of maximum dimension definition affects sample volume, number concentration, and all derived products such as the particle size distribution. For our in-situ imager B-ICI the sample volume is independent of the particle size for all sizes but the very smallest ice particles (collection efficiency drops for very small sizes, it is 80% at 20 µm and 50% at 12 µm size). Thus, our choice of maximum dimension does not affect accuracy of particle size distribution above approximately 20 µm. This is a further advantage of our sampling method. A discussion of this issue has been included in the improved description of the instrument.*

xiv.    **Page 4 Line 1** : Compact particle are spheroidal : ok but you might want to use spheroidal in the abstract to avoid any misunderstanding.

*Rephrased to:*
*"Compact particles have no pronounced features deviating from a compact geometry and include particles of spheroidal shape."*

**2.2.2 Radiosonde, LIDARs and RADAR-LIDAR**
I have the feeling that LIDAR and RADAR data could be more thoroughly exploited to complement the cirrus in situ measurements (in a discussion section for instance). As mentioned by the authors, those measurements can be used to describe the dynamical properties of the atmosphere. These additional measurements experiments would strengthen the main findings of this paper. In the present form of the paper, I don't really see the added value of such measurements (the lidar figure is not described and the radar figure needs a better description/analysis : see comment section 4 and figure 7)

*We prefer not to expand on the LIDAR results and interpretations would go beyond the scope of this article. Please, also see our response to "2. Major Comments" above and to "Page 4 Line 24" below. The figure (old Fig. 7 now new Fig. 4) has now been moved to Section 3.1 (weather condition) and is better explained.*

xv.    **Page 4 Lines 15-16** : "Radiosonde data, temperature, humidity, height and geographical coordinates can be assigned to each particle" : this sentence does not sound right. The use of the word "particle" is ambiguous. Do you mean cloud layer with a 60m vertical resolution?

*A temperature, height and humidity can be assigned to each individual ice particle. However, since some particles were measured at the same height at the same time, the temperature and humidity are also the same. Furthermore, the temperature and humidity do not change so quickly with altitude.*

*Slightly changed sentences:*

*"A radiosonde is connected to the in-situ imager. It measures temperature, humidity, altitude and geographical position. Thus, these parameters can be assigned to the photographed ice particles. "*

xvi.   **Page 4 Line 24 :**You should shortly sum up the main results of the in situ imager – Lidar extinction coefficient comparison. Otherwise, I don't understand the meaning of this sentence.

*Thanks for the advice. In the manuscript, this part has been expanded accordingly:*

*"The backscattered signal is used in this study as complementary information to assess the temporal and spatial characteristics of the ice clouds sampled with the in-situ imager. The extinction coefficients retrieved from LIDAR measurements compare favourably with the extinction measurements of the in-situ imager (Kuhn et al., 2017). The LIDAR beam and the balloon instrument probe the cloud at two locations close to each other. However, a certain distance remains resulting in an uncertainty when comparing extinction coefficients directly. An additional uncertainty arises from the fact that the LIDAR ratio (extinction coefficient/ backscatter coefficient) is not known. The in-situ data may help to constrain the LIDAR ratio, which will be tested in future with more joint data from our ongoing campaign."*

**3. Classification of measurements**
xvii.   **3.1 Cirrus origin**
xviii.   **Table 1 Page 5 and Line 11 Page 6** : Table 1 is interesting but I think average Temperature and Altitude values could also be mentioned here.

*Thanks for the advice, top and bottom of clouds as well as mean temperature are now listed in Table 1. The table is no longer sorted chronologically, but first the 4 in-situ origin days and then the 4 liquid origin days are listed.*

xix.   Could you also explain in the text which kind of weather maps and satellite images were used to describe the meteorological situation?

*Mostly ground pressure maps with front lines (DWD) & 500hPa  geo potential maps calculated from GFS model (accessed on www.wetter3.de),  MSG (Eumetsat) satellite image archive (http://www.woksat.info/wos.html)*

*"Weather conditions are analyzed using weather maps, such as ground pressure with frontal analysis (from DWD) and 500 hPa geopotential (accessed on www.wetter3.de), and IR satellite images (from MSG-Eumetsat accessed on http://www.woksat.info/wos.html)."*

xx.   **Figure 2 Page 6** : You mention latter in the text that the assignment between irregulars and rosettes was sometimes ambiguous. What about plate and compact spheroidal ice crystals?. Looking at figure 2, I can imagine that it is quite hard to discriminate small compact crystals from small plates. It looks like the shadow of the coating is distorted/modified by the impact of the ice crystal on the coating. It might

result in an increase of the degree of "roundness" of the ice crystal, meaning that if an automatic classification algorithm is used small ice plates could be classified as compact ice crystal (explaining that you find almost no plates in your cirrus cases). Am I wrong? Could you discuss mis classification issues?

*Particles were sorted into shape groups seven times independently of each other. Details of how much the sorting fluctuates are now given. The very small particles are problematic, because by sinking into the oil and the resulting shadow, some particle-edge features can appear rounder.*
*The subjective effect also has an influence. Even if the same person determines the particle shape, it varies from time to time.*
*For example, on the day with the most very small particles (2013-02-20) the values fluctuate as shown in the table below. The next table shows mean, max, min and standard deviation in percent of the particle shape frequency on 2013-02-20. Overall, the percentages of the frequency of each group deviate less than 5%.*
*With particles smaller than 20µm (corresponding to about 6% of all particles on 2013-02-20) the shape is hardly recognizable and compact and irregular are probably overrepresented. An automatic particle shape algorithm would have problems classifying the particles correctly. However, we do not use automatic classification, thus we are quite sure that plates are not misclassified, apart from the uncertainties related to smaller than 20-µm ice particles mentioned above. Nevertheless, we are working on an algorithm and testing this.*

| | Thomas 1 | Thomas 2 | Veronika 1 | Veronika 2 | Veronika 3 | Veronika 4 | Veronika 5 |
|---|---|---|---|---|---|---|---|
| Com | 63.44 | 75.99 | 67.03 | 65.59 | 63.44 | 73.48 | 71.33 |
| Irr | 16.13 | 20.79 | 23.66 | 15.77 | 19.71 | 16.85 | 12.19 |
| Col | 10.75 | 1.43 | 2.51 | 9.32 | 9.32 | 6.81 | 7.53 |
| Ros | 5.02 | 0.00 | 3.94 | 4.66 | 3.58 | 0.36 | 5.02 |
| Pla | 4.66 | 1.79 | 2.87 | 4.66 | 3.94 | 2.51 | 3.94 |

| | Compact | Irregular | rosettes | Plates | Columns |
|---|---|---|---|---|---|
| Mean | 68.6 | 17.9 | 3.5 | 3.2 | 6.8 |
| Std | 5.0 | 3.8 | 2.2 | 1.1 | 2.1 |
| Max | 76.0 | 23.7 | 5.0 | 4.7 | 10.8 |
| Min | 63.4 | 12.2 | 0.0 | 1.8 | 1.4 |

You should also show the size of the ice crystals on figure 2.
*100µm bar is placed on figure 2 now.*

xxi. **Page 6 Lines 4-5** : I think a verb is missing in this sentence, please consider rewriting this sentence.
*Now 'is greater than zero' is spelled out:*
*"… or 24 h before the in-situ measurement in case IWC was greater than zero during these 24 h."*

xxii.   **Page 6 Lines 6-10** : You might want to clarify this paragraph. I know that you are not supposed to fully describe the methodology described in Kramer et al., 2016 and Luebke et al., 2016. However, I think it is still necessary to elaborate on this cirrus classification as it is linked to the in situ microphysical properties.

*We have now described it more precisely.*

*"Consequently, the cirrus origin was determined here using temperature and IWC along 24 h back trajectories. The Lagrangian microphysical model CLaMS-Ice (Luebke et al., 2016) was used to calculate these trajectories, starting from locations along the balloon flight paths and using ECMWF ERA-Interim meteorological fields as input. Temperature was interpolated onto the trajectories, while the IWC along the trajectories was simulated with CLaMS-Ice . The origin of the observed cirrus cloud was identified as in-situ if the temperature of the trajectory was always below 235 K. In case the temperature was originally higher than 235 K and carries already ice water at the time temperature crosses 235 K towards colder values, the observed cirrus is assigned as liquid origin. The resulting classifications are listed in Tab. 1. Half of the measured cirrus clouds are classified as in-situ origin, the other half as liquid origin."*

**3.2 Weather conditions**

xxiii.   **Page 6 Line 13**: What are the average cloud heights?

*The mean height is the middle cloud height, i.e the arithmetic mean between the bottom and top heights of the cloud. In the text, the average height has now been replaced by middle height and explained the first time.*

xxiv.   **Page 7 Line 3** : I see that now the RADAR ESRAD is mentioned and used to detect the occurrence of Lee waves or gravity waves. For my personal understanding, could you explain me how this is done?

*We have now described better how this is done in the manuscript (Section 3.1, Page 9, Lines 2-5):*
*"The RADAR can yield vertical velocities based on the Doppler shift of the backscatter signal. The variation of vertical velocities over time and altitude shows very clearly that there were waves present at that time, horizontal wind direction points to the mountain range as source.*
*In the case of LIDAR, the extinction coefficient shows the appearance and disappearance of clouds and the slope of the clouds indicates waves."*

xxv.   **3.3 Cloud properties**

**Table 2 :** Table 2 is not easy to read and does not look very "attractive". But it is still quite important. I would recommend modifying it or maybe transforming it into a graph (if possible). If you want to keep that table, please use the same date format as the one used in table 1, use colours according to the air mass origin ( in accordance with figure 3).

*Table 2 has been removed. New graph with radiosonde profiles in Figure 5 shows temperature, relative humidity with respect to ice and cloud bottom and top, 4 liquid origin days in upper row and 4 in-situ origin days in lower row.*

*All NC are shown in Figure 8 and the mean NC is listed in former Tab 3 (now Tab. 2).
Average particle sizes, Min and Max were added to the former Table 3 (now Tab. 2).
Date formats are now the same, furthermore the table is no longer chronological, but first
the in-situ cases and then the liquid cases are listed.*

xxvi.     **Page7 Lines 7-10 :**I'm getting lost here, I don't understand how a cirrus could have a
          geometrical thickness of 6km and a cloud base close to 2km (and temperature of -
          11.5°C). Could you elaborate on the cirrus definition used in your study ?

*In these cases, the clouds are completely frozen, previously mixed phase in the lower part
of the cloud and cirrus in the upper part of the cloud. These layers are not separable.*

These two thick clouds have a liquid origin and are associated with southerly winds. Looking
at Kramer et al., ACP 2016, and Luebke et al. 2016 I can read that liquid origin cirrus are
characterized by :

(1) high IWC, high ice crystal concentration (NC>100 L-1), and large ice crystals (D>200µm)
*Both values, NC and IWC, are on average smaller in  liquid origin clouds. The liquid origin
cirrus formed at lower altitudes and temperatures above 235 K, where typically mixed-
phase clouds occur. They are uplifted into the in-situ temperature range where they at
latest fully glaciate. In the original altitude, more water vapor and INPs are available,
resulting –together with the continuous updraft--  in larger particles, higher number
concentration and thus higher ice water content compared to the in-situ origin clouds.
Such type of clouds are present typically in case of convection or large scale transport like
warm conveyor belts.*
(2) nucleation mechanism is probably homogeneous freezing (low IN)
*Nucleation mechanism is most probably: initially heterogeneous, maybe followed by a
second homogeneous freezing event. In the slow updrafts in frontal systems, also
homogeneous freezing does not produce a high ice crystal number.*
(3) Fast updrafts :
*Liquid origin cirrus are present not only in fast updrafts, but also in slow updraft systems
like the frontal systems observed here.*
(4) They appear with liquid containing clouds below
*In our case the liquid containing, mixed phase clouds below are already completely frozen.*
From your results presented in table 2, we can see that the ice crystal size is on average
larger for liquid origin cirrus but the ice number concentration is very low (especially for the
01.04.2015 & 12.02.2016 case). How do you explain this? It doesn't not seem to agree with
mid latitude results presented in Kramer et al., 2016 and Luebke et al., 2016.

*The mid-lat liquid origin observations presented in Krämer et al. 2016 and Luebke et al.
2016 show higher NC than observed here – it is explained in the last paragraph in section
4.1 (Size and number concentration) that the reason is most probably the lower INP
number in the Arctic.
In contrast, the mid-lat in-situ origin observations show lower NC than those in the Arctic.
The reason is that most of the Arctic in-situ observations are influenced by mountain
waves with high vertical velocities triggering homogeneous nucleation of many ice
crystals. Such observations were very rare in Krämer et al. 2016 and Luebke et al. 2016.*

*However, in Krämer et al. 2016 two types of in-situ cirrus are discussed, namely slow and fast updraft in-situ cirrus, where the fast updraft are explained to appear i.e. in mountain waves and have high ice crystal numbers.*
*This can also be seen in the new article of Gryspeerdt et al. (2018), ACPD where a map of cirrus NC (derived from satellite observations) is shown (their Fig. 1 b).*
*So, there is no disagreement with Krämer et al. 2016 and Luebke et al. 2016.*

I'm also wondering if the low layers considered as cirrus clouds correspond to mixed phase clouds, glaciated clouds or fall streaks?

*The two thick clouds had completely frozen mixed phase clouds at their bottom without a gap to the cirrus above (the layers are not clearly separated from each other). Thus, these layers at the bottom of the cloud may previously have contained liquid drops, as observed by Kramer et al., ACP 2016 and Luebke et al. 2016 for the mid-latitudes.*

How can you tell that low level cloud layers are solely composed of ice crystals : you have no cloud droplet measurements ?

*With our instruments we can differentiate between frozen particles and liquid cloud drops. In our measurements here, all particles were apparently frozen, and we are quite confident in this differentiating between ice and liquid based on experience. In fact, we have detected liquid layers in Lindenberg (Germany) at a temperature between -10 and -20°C (Wolf et al. 2017, Geophysical Research Abstracts, Vol. 19, EGU2017-7708, 2017, EGU General Assembly 2017).*

xxvii. **Page 10 - Table 3** : Table 3 displays the distribution of ice crystal habits within each "flights". It is interesting but hard to compare. An indication of the temperature and relative humidity with respect to ice should be provided along these values. A vertical distribution of the cloud shape would also be more valuable.

*Figure 5, which shows the radiosonde data, has been added to the manuscript.*

*We have looked for each day at the particle shape occurrence evaluated in various temperature ranges. From that we could not see a clear temperature dependence of shape occurrence. Since the temperature decreases monotonically with height, this also means that there was no apparent height dependence. Rather, the particles are evenly distributed over the different layers.*

*The table below shows the average temperature at which the corresponding particles have been collected. These averaged temperatures show only minor differences. In the case of in-situ all particles were collected between -40 and <-60°C. Liquid origin particles were collected between -10 and <-60°C.   In 3 of 4 in-situ origin cases and in a liquid-origin case, the compact particles were found at slightly lower temperature. With decreasing temperature (increasing height) the particles are rather smaller and therefore more (Kuhn 2016).*

*A dependence of the shape from the humidity is not recognized.*

|  | Compact | Irregular | Columns | Rosettes | Plates |
|---|---|---|---|---|---|
| 2012-04-04 | -53.9 | -51.2 | -52.5 | -50.6 | -51.8 |
| 2013-02-20 | -63.6 | -60.5 | -62.4 | -61.7 | -60.5 |
| 2016-03-15 | -57.2 | -55.9 | -53.1 | -50.2 | -64.3 |
| 2016-12-15 | -65.3 | -64.9 | -65.3 | -64.3 | No plates |
| 2013-12-18 | -53.0 | -52.5 | -52.5 | -52.5 | -52.5 |
| 2014-03-20 | -53.4 | -50.0 | -50.2 | -50.0 | -54.2 |
| 2015-04-01 | -48.1 | -46.6 | -45.6 | -49.9 | -43.1 |
| 2016-02-12 | -38.0 | -40.3 | -41.0 | -37.6 | -39.2 |

In your statistics you are "mixing" ice crystals measured at 2000m/-11°C with ice crystals found at 8000m/-54°C and compare it to ice crystals found at 11km/-65°C ? Is this relevant ?

*Since we have not found a temperature dependence of shape occurrence within a cloud, we think it is ok to compare the ice particles from these different layers.*
*In these cases of thick clouds, where the lower part of the cloud is a previously mixed-phase cloud, the ice particles are of liquid origin, as in the cirrus part above. Together with the differences we have found in shape occurrence for the two different cloud origins, this confirms that the cloud origin is more important, i.e. that temperature plays a major role in the formation of the clouds. This seems to be more important than temperature variations within the same cloud later on.*

In *in situ* cirrus, the fraction of compact ice crystals seems to be high (40% to 70%). Is this in agreement with previous results found in cirrus clouds?
The fraction of plate is very low but don't you think it is due to a possible misclassification of small plates to compact ice crystals. Once again, this should be discussed in the paper.

*Others (Korolev 1999) have seen only a few plates and columns. Yes, misclassification can occur, especially the smaller the particles are. As already mentioned, we can determine the particle shape from 20μm with quite good certainty.  On 2013-02-20 about 6% of all particles were smaller than 20μm. If all these small particles would be plates, then the frequency of occurence of plates would be higher by 6%. This can be seen as an upper limit of uncertainty for plates.*

*The low number of plates probably corresponds more closely to the temperature of the cloud and the formation of ice particles.*

**4. Results and Discussion**

**4.1. Size and number concentration**
xxviii.    **Page 10 line 3** : "see observations 2" : what does it mean ? Maximum size displayed on table 2.
*Yes, Tab. 2 is meant. This has been corrected in the manuscript.*

xxix.   **Page 10 line 5** : "At three of the four days" should be something like "During three of the four days"

*Changed to "On three of the four days…"*

xxx.   **Figure 5 – Page 12** : I think that you should show your results in log-log scale (with dN/dlogDmaxvsDmax for instance) – not mandatory as you might not see the difference (broadness of PSD) highlighted in the paper.

*Thank you for the suggestion, the features of the PSDs can be clearly seen in log-log scale. We are showing our results now on log-log scale. We have chosen to keep the PSDs as dN/dDmax. The normalization dN/dlogDmax, which is more common for aerosol PSDs, would require a re-normalization of our data. We have checked the features of the PSDs in both dN/dDmax and dN/dlogDmax, and they appear strong and very similar in both normalizations. In Sect. 2.2 Measurement methods, we explain which normalization we are using in the PSDs to avoid any misunderstandings.*

However, I think an additional panel where the PSD measured at comparable temperature should also be shown. It would help support your main conclusions regarding the differences of PSD behaviour found for liquid origin cirrus and in situ cirrus.

*Fig 7. (new) shows now PSD for all measurement cases, where possible also at different cloud levels.*

xxxi.   **Page 10 Lines 11-15** : It would be good if you could rephrase this paragraph to help the reader understand your point. "vastly" should be significantly.

*The complete size and number concentration part has been rewritten*

The fact that the PSD is narrower with increasing height and decreasing temperature is clearly evidenced on the in situ cirrus case. Size is decreasing and NC is increasing. The PSD is very narrow and almost look like monodispersed distribution, is it really representative?

*For all 4 in-situ origin cases the PSD is narrow. In the case of liquid origin cases wider. The in-situ PSDs corresponds to quite young homogeneously formed cirrus, where the ice crystals have not yet grown to larger sizes.*

Is it due to sampling issues?

*We cannot think of a way this could be a sampling issue. With the improved descriptions in Sect 2.2 (Measurement methods) this should also be clearer to the reader.*

This temperature/altitude trend is not clearly seen for the liquid origin cirrus case. Why ? Do you have microphysical process hypothesis to explain this behaviour?

*We think that in the in-situ origin cirrus the altitude/temperature structure is caused by the main ice nucleation zone being at the cloud top (coldest point). This structure is also found recently in observations on a global scale (Gryspeerdt et al., 2018, ACPD, Fig. 2, upper row).*
*Liquid origin cirrus do not form at the cloud tops, but at lower altitudes and then ascent in the prevailing updraft. The change in PSDs while ascending is caused by loss of ice crystals on their way up, mostly of large crystals due to sedimentation.*

xxxii. **Page 10 Line 16** : "While these differences are obviously not related to local ambient conditions, they are related to the cloud origin" : this statement might be a bit strong. Without showing additonnal cases, it is hard to be so positive... What about humidity measurements? I did not see any in the paper. It could be useful to better interpret your dataset.

*Sentence slightly softened. Relative humidity was now also considered.*

*"While these differences are obviously not only related to the local ambient conditions, they are strongly related to the cloud origin."*

xxxiii. **Page 10 Lines 17-18** :Gayet al., 2007 focused on a case study where observations of ice crystals precipitation (from cirrus ?) down to a supercooled boundary layer stratocumulus were made. Measurements were performed at 1500m/-11°C. The PSD shows ice crystals with size ranging from 25µm to 1000µm with a Deff=270µm (and NC=10 l-1). I understand that in situ measurements in arctic cirrus are scarce but this study is hardly comparable to your study. At least you need to be more precise in comparing your results, do you mean that you are comparing the PSD of precipitating ice crystals (which case is this in your study ?) to Gayet et al., work ?

*We agree with you that Gayet described a measurement in which they collected falling ice particles from cirrus clouds. However, the size distribution between 25µm and 1000 µm mentioned by them corresponds well with our two thick liquid origin clouds, which had their lower edge approximately at the same height and similar temperature.*

*Now in manuscript:*
*"Gayet 2007" described a measurement in which they collected falling ice particles from cirrus clouds. The size distribution between 25µm and 1000 µm mentioned by them corresponds well with our two thick liquid origin clouds, which had their lower edge approximately at the same height and similar temperature."*

*A new paper with PSDs is Sourdeval et al (2017), ACPD. It does not separate between in-situ and liquid origin, but it is discussed that the mode in the PSD originates from liquid origin cirrus for large ice crystals that only appear at temperatures > -50C. However, there are no Arctic measurements there - a comparison with our measurements shows that in the Arctic liquid origin clouds with larger particles can still occur at lower temperatures.*

xxxiv.    **Page 10 Lines 21-23** Yes, I agree that the number concentration of ice crystals found in this in situ cirrus is higher than in the liquid origin cirrus. This is not in agreement with previous findings of Kramer et al. and Luebke et al..

*As already explained in the answer to comment Page 7 Lines 7-10, it is in agreement.*

I think that all your cases should be presented on Figure 6. It would be easier to see if the vertical profiles are linked to the in situ/liquid origin or the air mass origin. It is hard to draw conclusions based on two very specific cases.

*All NC are now shown in Figure 8 (new). Two of the in-situ cases have very high NC, all other measurements have a quite similar NC.*

xxxv.    **Page10 Lines 23-24** : "It should be noted that the y axis …. in concentration" : you could delete this sentence.

*Sentence deleted*

xxxvi.    **Page 11 Lines 1-5 :**Fig 6 is very important but I don't understand why only two cases are shown. If possible, the 8 flights should be plotted on this figure. You also say that two cases (half of your in situ cirrus events) of in situ origin cirrus cloud (20/02/2013 & 15/03/2016) exhibit high ice crystal number concentrations, sometimes much higher than concentration found in liquid-origin cirrus. It is true for the 20/02/2013 case but I don't think this the case for the 15/03/2016 where concentration is close to 11-14 l$^{-1}$ on average (according to table 2). Some cases of liquid-origin cirrus reach 56 l$^{-1}$ and the 04/04/2012 in situ origin cirrus concentration reaches 131 l$^{-1}$ at 7km. So, I don't understand your comparison. Please, clarify this point as it does not make sense to me. Once again, this also shows that each profile should be plotted on this figure to facilitate the comparison and draw solid
conclusions.

*There was a mistake in the manuscript: 'two measurements (20.2.2013 and 15.3.2016)' should have been 'two measurements (20.2.2013 and 4.4.2012)'.*
*All NC are now shown in Figure 8 (new). Two of the in-situ cases have very high NC (20.2.2013 and 4.4.2012) the 15.3.2016 case has similar NC as liquid origin cirrus; on 15.12.2016 the NC was extremely low. All liquid- origin measurements have quite similar NC.*

xxxvii.    **Page 11 – Lines 9-11 and figure 7** : It is a good idea to use lidar and radar measurements but I think that you need to go more into details. You show the vertical profile of the extinction coefficient measured from the LIDAR but I don't see the added value of such plot : nothing is said about it or compared (extinction, altitude, structure of the cloud...). What about the lidar and measurements performed during the liquid-origin cirrus event?

*We have moved this sentence and image to section 3.2. weather condition. In this article we are using LIDAR and RADAR as help for finding waves.*

*Please, also see our response to "2. Major Comments" and to "Page 4 Line 24".*

xxxviii.    **Page 11 – Lines 10-11 figure 7** : Without a more detailed explanation it is hard to see/understand how wind vertical velocity measurements below 5km can explain "waves with high velocities can explain such higher number concentration". Please clarify this.

*The sentence as it was in the manuscript "waves with the related high vertical velocities can explain such higher number concentration" refers to the fact that vertical velocities lead to adiabatic expansion and contraction of the vertically moving air parcels with the resulting cooling and warming at certain locations. And that means that water saturation pressures also change. It should also be noted that vertical velocity is the driver of high (homogeneously nucleated) ice crystal concentrations (Kärcher and Lohmann, 2002).*

*Here also two references for basics on such waves:*
*David C. Fritts and M. Joan Alexander: Gravity Wave Dynamics and Effects in the Middle Atmosphere, Review of Geophysics, 2003, Vol. 41, p.3-1 -- 3-64.*
*Carmen J. Nappo: An Introduction to Atmospheric Gravity Waves. Academic Press, 2012, 359pp.*

*Sentence changed to: "These gravity or mountain lee waves with the related high vertical velocities can be the needed trigger for such higher number concentrations."*

xxxix.    **Page 11 -Lines 14-16** : This could be an explanation, indeed. From your results, one can see that the ice crystal sizes agree with Luebke et al. But not the concentrations. The reasons for such discrepancies should be discussed and your results should be compared to other measurements in cirrus clouds (at mid latitude and in the Arctic if there were any).

*It has been discussed and is now even more extensive.*
*Furthermore, it is not possible to compare this type of measurement with others in the Arctic. In our opinion, this is the first time that the properties of ice particles of Arctic cirrus clouds have been studied according to their origin.*

 I also have the feeling that the vertical distribution of Nc is much more variable for in situ origin cirrus than for liquid origin cirrus, why ?

*Yes, the NC in the case of in-situ origin is more variable, because with the in-situ origin cirrus the NC can be strongly modulated by the variability of the vertical speed. With liquid origin, NC usually depends on the number of INPs that are less variable.*

Don't you think it is a problem to compare cirrus properties at very different altitudes ?
 I think that you sometimes compare fall streaks, high and cold cirrus (-66°C-10000m), with warm low ice clouds (-11.5°C -2000m ) ?

***No, we don't think so. Now we show all the data and discuss the results. Please, also see
our responses to comments XXVI and XXVII.***

xl.     **Page 11 – Line 16** : should be "Arctic region"

***Changed, thank you***

**4.2 Shape**
xli.    **Page 11 Lines 20-25** : This paragraph is more a discussion than actual results. It
        should be moved either to a new discussion section or to line 10 p 12. Your
        paragraph should start with "The frequency of occurrence of the different particle
        shape... line 26.

***Order has been changed.***

xlii.   **Page 12 Line 6** : "this corroborates  findings by others" : which findings ? be more
        specific.

***"this" referred to the previous sentence. Now the sentence has been changed to:***

***"This corroborates findings by others (e.g. Weickmann et al., 1948; Heymsfield et al., 2002;
Schmitt et al., 2006), in which measurements showed that around 80% of all collected
rosettes were hollow to a certain extent."***

It is important to compare your results with other measurements. For instance, I am
surprised to see that rosettes are mainly found in liquid-origin cirrus, at which temperature?
. My question is : Do you really think that the shape of the ice crystals is more likely to be
influenced by the origin of the cirrus (meaning in situ or liquid) or the temperature and Rhi ?

***Our measurements do not really show a vertical distribution of particle shapes as a
function of temperature. But if you compare the temperature range (-70 to -10°C) and the
mostly quite low supersaturation (up to max 130% relative humidity over ice) with the
diagram by Bailey 2009, you can see that over this whole range a group of "compact
faceted polycrystals, thick plates, occasional short columns and equiaxed" exists.
Furthermore, most of the particles in the measured temperature range can be assigned to
the polycrystalline and columnar regime. According to Bailey, more supersaturation is
needed for the growth of rosettes. The supersaturation present in our measurements is
usually too low for rosette growth. This suggests that the particles have formed and been
advected beforehand.  This is probably more the case for the liquid origin clouds from the
south, as it was warmer in those and therefore more water vapour can be in the
atmosphere.***

***Furthermore, it is important that in the temperature range in which in-situ cirrus form (< -
38°C) the water concentration in the atmosphere is significantly lower than at the
temperatures at which the liquid origin cirrus form (> -38C; the temperature at which they
are detected is then colder because they have risen).  Therefore there is simply less water
available to form complicated shapes, the colder the less.***

*It is also important to remember that it is completely new to associate the shapes with the origin of the cirrus.*

xliii.   **Page 12 Lines 5-10** : please rephrase this paragraph, I don't understand what you are trying to show.

*Text in this section has been in part re-arranged and re-phrased in the manuscript to make the discussion clearer.*

**5. Conclusions**

xliv.   **Page 13 Line 7** : "when looking at the cirrus in terms of its origin, similarities between the various properties are striking" : I don't understand what you mean here : you are saying just above that large differences in ice particle size, shape and number are observed and then that similarities are striking when looking at the origin of cirrus.... please rephrase.

*Sentence has been rephrased.*

*…are expected to vary in accordance to cloud origin. And indeed, while large differences in particle size, shape and number concentration are observed between the various measurements, some similarities are noticed within the two groups of data with liquid and in-situ origin clouds, respectively. These similarities and the differences between data, when grouped in liquid and in-situ origin, are summarized below:*
*1)…*
*2)…*
*3)…*
*The results of this study imply that remote sensing …*

xlv.   **Line 8-9 :** I think this sentence should be placed after the summary of the most important results.

*As proposed, the sentence is now after the list of the main results.*

xlvi.   **Page 14** : I would suggest to also summarize the comparison between your work and previous studies using the same cirrus classification.

*Until now, hardly anyone used this new classification, except Luebke et al. 2016. Wernli 2016 discussed the frequencies of in-situ and liquid origin cirrus. In the last point (number concentration) the comparison with the Mid-Lats was added.*
*"In comparison, lower number concentrations were measured in the mid-latitudes for this cloud type, as hardly any wave-induced in-situ origin clouds were observed."*

*"In contrast, high number concentrations were measured in the mid-latitudes for this cloud type, as there is a higher number of INPs in the mid-latitudes than in the Arctic"*

Review of Ice particle properties of Arctic cirrus by Veronika Wolf et al.

General comment:

In this study, arctic cirrus clouds are investigated, using measurements from balloon-borne instruments. The data from eight radiosonde ascents are investigated about shape, size and number concentration of ice particles. In combination with trajectory calculations, the formation pathway can be determined and the microphysical properties can be related to these pathways.

Overall, this is an interesting study using a very promising technique for the detection of ice particles on a very well suited platform; thus, this is an adequate and meaningful contribution to ACP. However, there are some issues which should be clarified before the manuscript can be accepted for publication. Therefore I recommend major revisions for the manuscript.

***Thank you very much for this evaluation and the comments***

In the following I will explain my concerns in detail.

Major points

1. Definition of liquid origin and in situ formation not clear The study relies strongly on the recent developed classification scheme by Krämer et al. (2016), separating ice crystal formation pathways into liquid origin and in situ formed ice crystals. However, the definitions of these two types seem not to be correct from a thermodynamic point of view: liquid origin is characterised by formation at water saturation, while in situ formation occurs at conditions below water saturation. Please correct and extend the definitions in the manuscript accordingly, see also Krämer et al. (2016) or even Wernli et al. (2016).

***This comment has been answered in our response to the comment "xxii. page 6 lines 6-10" of Referee 1. The definition has now been described in more detail as explained in that response.***

2. Interpretation of data and scientific results While the measurements of the ice crystals show very high quality and seem to be quite interesting, the evaluation of the data is weak. It is not really, what the authors want to state with their results. Especially, the interpretation of the data concerning the different pathways is not clear. What is the story you want to tell? What did you expect for ice crystal shape, size and number concentrations for the different formation mechanisms? What is the result and how can this be interpreted? Is there any hint from theory to corroborate these findings (was it expected or surprising, and why?)? Invest more theory for the interpretation of the data and the presentation of the results. Finally, it would be nice to have figures of the profiles, at least in the appendix.

*The discussion has been significantly expanded while responding to the comments of Referee 1. This should cover the interesting questions that you have raised.*

Minor points:

1. High speed measurements: Actually, high speed measurements have some other issues beside the problem of shattering, see e.g. the compression of air as indicated in the study by Weigel et al. (2016).

*Thank you for pointing that out. The article now refers to this problem.*

*Another problem with aircraft measurements, described by Weigel 2016, is that the air around the wing under which the instrument is mounted is compressed. As a result, in order to calculate the NC, the temperature and pressure must be corrected to match the ambient conditions (undisturbed).*

2. Classification of data partly manually/automatically: It is stated in the text, that the classification was carried out partly automatically. Please describe how this was done and which techniques were used.

*In Section 2.3.2 (Image processing), the image analysis is now described in detail.*

3. Measurements with RADAR/LIDAR: What was the outcome of the complementary measurements of RADAR and LIDAR? Is there any additional value for the results/interpretation?

*The extinction coefficient obtained from LIDAR matches well with that of the in-situ imager (Kuhn2017). In a future article, when we will have more joint data from LIDAR and our balloon-borne in-situ imaging, we will compare the particle shape with the depolarization ratio. See also the responses to comments related to LIDAR and RADAR measurements by Referee 1.*

4. Listing of the different clouds in table 2: It is not clear to me, how the authors can count 4 clouds, because it seems that there are two adjacent layers, since the top layer of the first cloud (e.g. 5680m) is the same as the bottom layer of the next cloud. Please explain this interpretation.

   *Table 2 has been removed and some of its information added to Tab. 1 and Tab. 3 (now Tab. 2). We measured on eight different days. On each measuring day there was a layer of clouds (sometimes very thick, sometimes very thin). If possible, we have divided this cloud into different layers, for example to obtain PSDs at different heights.*

*This is the response to Short Comment SC1.*
Overall: This manuscript needs to be improved significantly. There are many issues related to text flow and scientific understanding of the Arctic cirrus clouds, check on cirrus dynamics from SHEBA project. HEBA project: https://link.springer.com/content/pdf/10.1007%2Fs00703-003-0009-z.pdf Results are also contradictory for the theory of parameterizations and needs to be clarified. More cases wrt satellite and lidar/radar should be used and connected to IC concentrations. Presently content is poorly written and not discussed based on other works in the Arctic clouds. Specifically, liquid origin and local origin concepts are misleading formation of these clouds. There are many issues with this paper and they are listed as:

*Thank you for the comments. We have tried to understand these, however, we suspect that the article was misunderstood in a few points. We hope that, with the answers to the two referees and the improved manuscript, all questions are answered and misunderstandings can be solved.*

1. abstract is not given explicitly; no info on what kind of balloon being used?

*For all measurements, a plastic foil stratospheric balloon filled with helium was used.*

2. what sensors are used?

*Instruments used are the in-situ imager, described in Kuhn 2013 and a radiosonde RS92.*

*If possible, the measurements were supported by LIDAR and ESRAD.*

3. no meaning of liquid clouds at cirrus level? Not good naming, and very confusing.

*Liquid origin cirrus is not equal to liquid cloud. Krämer 2016 and Wernli 2016 have described two new types of cirrus origin, which are dependent on temperature, IWC and vertical wind. Liquid origin cirrus is cirrus which was formed via the liquid phase. I would not want to change the name, as these authors have coined these terms since 2016.*

4. in-situ origin cloud? Cirrus form due to IN and its properties are related to local or advection.

*This is described in part 3.2 (Cirrus Origin), for more information we recommend reading Krämer 2016, Wernli 2016 or Luebke 2016.*

5. how do you explain the liquid origin and local origin? This doesn't make sense; and you don't have a mechanism to explain it.

*See response to comment 4*

6. 61% compact??? And 25% irregular, is this a resolution issue? Seems to me it is resolution issue unless you have a proof of it.

*The resolution of the images is very high 1px =1.65 mu. The procedure is described in Sect. 2.3.2 Image processing. Problems with sorting are pointed out. See also answer to comment "Figure 2 Page 6" by Referee 1.*

7. page2; no shattering at this level because already they are small, take out refs on this. Balloon is not like airplane.

*This is true for the very small particles in higher altitudes or in the case of in-situ cirrus clouds. But with the liquid origin cirrus, at lower heights, there are partly very large particles (>500mu), which could have been fragmented when using the older devices.*

8. what parameterizations?

*Parameterizations of ice particles microphysical properties like size, shape number concentration*

9. "we detect particles. . ..." no you don't, sensor does.

*Sentence was rephrased*

10. depends on ambient conditions. . .. . . do not include waves, systems, and temperature together. . ... Confusing and not meaningful. What is role of T wrt waves or systems. Talk about its physics, T ok.

*See reply to Referee 1 comment "Page 2 – Lines 25-30"*

11. For these reasons????? What reasons?

*Sentence was rephrased*

12. introduction is confusing and not clear.

*Introduction is improved and the order changed slightly*

13. location; what level (height) measurements were taken? Is this cirrus or arctic BL cloud?

*Clouds between 3-12 km. The lower cloud layers can count as completely frozen, previoulsy mixed-phase clouds the rest is cirrus.*

14; what is the in-situ imager? Imager of what? name should be ice crystal imaging probe or similar..... ICIP???? Check your earlier works, it says differently.

*It is another instrument and had no name so far. We have now called it Balloon-borne Ice Cloud particle Imager (B-ICI).*

2.2.1 In-situ imager

15. what is the compact means? I feel these are not resolved particles, out of focus particles.

*In new manuscript: "Compact particles have no pronounced features deviating from a compact geometry and include particles of spheroidal shape."*
*Kuhn 2013: "The optics is focused slightly above the film strip so that all particles will be in focus."*

16. page 4; lidar extinction? You should include some work here on this.

*See reply to Referee 1 comment: Page 4 Line 24*

17. radar and lidar images were not clearly used to support cirrus dynamics. But they should. Not enough to say water origin or local origin.

*See reply to Referee 1 comment Page 11 – Lines 9-11 and figure 7*

Table 1 should state height levels.

*Height levels are listed now*

Figure 2; size of these particles should be in the image. Again, what is the meaning of compact?

*Size mark added, compact see reply above*

Page 6; shows how did you use satellite images, show a case.

*We don't think this is so important for the content of the article.*

Page 7; smaller particles are not efficiently sampled. . ... how small?

*As described less than 10μm*

Page 8; Table 2; at >-60C, you have more IN, why you have these??? But not always true? It is against IN parameterizations, explain it.

*It is now better explained*

Fig 4; liquid origin? How do you know?

*Described in section 3.2 (Cirrus origin)*

Page 10; higher than this in liquid origin? Why? This is against the nature of formation again.

*It is now better explained*

Figure 5; what is the uncertainty in $N_i$ measurements? And what is the time period for collection of $N_i$? How did you calculate $N_i$?

*It is now explained in section 2.3.*

Figure 6; this figure useless; need to show sampling time, and number of points used in $N_i$ calculations. Need to show all other cases. $N_i$ is calculated what? TAS? Sampling area? Etc.

*New Figure 8 now shows all NC*

Fig. 7; you need to show calculation of ext here. Also you need to show at least cases with extreme conditions such as $N_i \sim 5$ and $N_i \sim 300$ L-1, and then discuss it.

*See reply to Referee 1  comment: Page 7 Line 3*

Fig. 7b; why the $V_d$ given at the BL is important for cirrus level? Don't you have a figure for cirrus level? You need a comparison table or figure for outcome of this work. Then explain what the results are significantly different.

*See reply to Referee 1  comment: Page 7 Line 3*

---

## Referee Report (RR1)

Review of revised version of
**Ice particle properties of Arctic cirrus**
by Veronika Wolf et al.

**General comment:**
The authors have mostly addressed my concerns adequately; the manuscript has improved a lot, constituting a good contribution to ACP. However, there are some minor issues which should be clarified before the manuscript can be accepted for publication. Therefore I recommend minor revisions for the manuscript.

In the following I will explain my concerns in detail.

**Minor points:**

- page 3, line 28 (and in the whole text):
  Why do you distinguish between "orographically induced gravity waves" and "mountain lee waves"? To my opinion, both denote the same phenomenon, i.e. gravity waves induced by stratified flow over (steep) topography (i.e. mountains). Maybe you can explain the term in the beginning and later just name them "gravity waves"?

- page 6, line 18/19:
  How was the LIDAR used? In any case, if available or for a spatial distance/temporal displacement between balloon and LIDAR below a certain threshold? Please explain this in more details.

- page 9, line 14:
  The defition of liquid origin vs. in situ ice clouds is still not complete (or even correct), especially the thermodynamic aspect (at approx. water saturation vs. below water saturation) is not mentioned. My suggestion would be:

  1. in situ: homogeneous freezing of solution droplets or heterogeneous nucleation (deposition nucleation) for $RH < 1$ and $T < 235\,\mathrm{K}$

  2. liquid origin: homogeneous/heterogeneous freezing of pre-existing cloud droplets for $RH \sim 1$ and $T > 235\,\mathrm{K}$.

  Please expand the text in this respect.

- Figures 5,7,8:
  For a better representation, you could use 2 blocks of $2 \times 2$ images for each ice cloud category.

- page 15, line 33 and following text:
  I don't think this explanation is completely correct. Even for very low temperatures ice crystals can grow to larger sizes and complex shapes (see e.g. fig. 5, lower panel, in Bailey and Hallett, 2009), if they get enough water vapour. The main issue here is the competition for water vapour, i.e. if many ice crystals compete for the available water vapour, they cannot grow to large/complex sizes. This is probably the case for 2013-02-20, since the number concentration of ice crystals is huge. Please change the text accordingly.

**Technical comments**
Figure 9, left panel: wrong key, it should read "2016-02-12".

---

## Author Response (AR2)

**Report #1**

The authors did a very good job and have gone to substantial effort to address my concerns, Thank you for taking into account my comments.

**Thank you very much for your comments, they helped us to improved the article.**
**In section 4.1, some explanations on the processes (dynamical-microphysical) responsible for the observed differences of particle concentrations seem a bit redundant. I encourage the author to go through this part again to improve the readability of the section.**

**Thank you for the advice. Section 4.1. has been revised and repetitions removed.**

The editor and myself are also a bit puzzled by the presumption that cirrus particles are passive tracers: if they came from sufficiently warm temperatures they must be liquid. Could you elaborate on this, don't cirrus clouds continuously regenerate themselves?

**We think that our explanation was a little bit misleading. We actually do not presume cirrus particles as passive tracers. It is more the cloud itself with certain microphysical properties (NC and IWC), which can be traced back to different origins.**

**Liquid origin clouds are typically clouds, which are generated at temperatures far above 235K. Associated to such clouds are for example warm conveyor belts, which transport moist tropospheric air masses from low level to upper tropospheric levels. Due to this strong uplift, clouds form first in lower levels and are continuously transported to lower temperatures up into the cirrus temperature range (< 235K) (see e.g. Wernli et al., JRL, 2016, Krämer et al., ACP, 2016). Ice particles are able to grow to large sizes due to the high amount of available water vapour in combination with the continuous cooling over a large altitude range. One could also say that in case of a liquid origin cirrus, there exist ice particles that formed at a temperature above 235 K.**

**In contrast, in-situ formed cirrus clouds do typically not experience lifting over a very large altitude and thus temperature range. In addition, in the upper troposphere at temperatures below 235K, there is not much water vapour available to grow the particles to very large sizes. One could also say that in the case of an in-situ origin cloud, there exist only ice particles that have formed at a temperature of or below 235 K.**

**Due to these differences in the air mass dynamics connected to cloud formation, it is expected that there are clear differences in microphysical properties between in-situ and liquid origin cirrus, which is already found in aircraft in-situ observations (Luebke et al., ACP, 2016).**

**This general microphysical difference between in-situ and liquid origin cirrus cloud could be less pronounced at the top of the liquid origin, where the updraft is not that strong anymore and the amount of water vapour is already reduced. In addition, sometimes there is also the possibility of in-situ formed cirrus on top of a liquid origin cloud due to slow uplift of air masses (Wernli et al, JRL, 2016).**

**While sedimentation and new ice formation may occur during the evolution of clouds and change certain cloud properties such as vertical distribution of ice, these processes will not change much the general difference between liquid and in-situ origin clouds. A liquid origin cloud would still contain ice particles that have formed at temperatures above 235 K, and an in-situ origin cloud would in any case only contain ice particles that have formed at temperatures below about 235 K. Thus, the presence of ice particles that formed at above 235 K could be regarded as some sort of "tracers" indicating liquid origin. However, this is not what we do in this study, instead we look at the air mass trajectories and at which temperature ice forms.**

**Report #2**

Review of revised version of

Ice particle properties of Arctic cirrus

by Veronika Wolf et al.

General comment:

The authors have mostly addressed my concerns adequately; the manuscript has improved a lot, constituting a good contribution to ACP. However, there are some minor issues which should be clarified before the manuscript can be accepted for publication. Therefore I recommend minor revisions for the manuscript.

**Thank you very much for your comments, addressing them improved the article.**

In the following I will explain my concerns in detail.

Minor points:

- page 3, line 28 (and in the whole text):Why do you distinguish between "orographically induced gravity waves" and "mountain lee waves"? To my opinion, both denote the same phenomenon, i.e. gravity waves induced by stratified flow over (steep) topography (i.e. mountains).

Maybe you can explain the term in the beginning and later just name them "gravity waves"?

**Thank you, for this remark. After explaining the phenomenon, we only refer to "waves" now.**

- page 6, line 18/19:How was the LIDAR used? In any case, if available or for a spatial distance/temporal displacement between balloon and LIDAR below a certain threshold? Please explain this in more details.

**We used the LIDAR in any case if it was available. To make this clear we have changed the following text:**

*From:*

*"Whenever LIDAR measurements are possible, these are used to complement the in-situ balloon-borne measurements. One LIDAR is located at the Swedish Institute of Space Physics (IRF) (about 30 km away from ESRANGE) and another one is located at ESRANGE close to the balloon launch pad. The LIDAR at IRF (Voelger 2005) is an elastic backscatter LIDAR and at ESRANGE a Raman-Mie LIDAR (Blum 2005). The backscattered signal is used in this study as complementary information to assess the temporal and spatial characteristics of the ice clouds sampled with B-ICI. The extinction coefficients retrieved from LIDAR measurements compare favourably with the extinction measurements of B-ICI (Kuhn 2017). The LIDAR beam and B-ICI sample the cloud at two locations close to each other. However, a certain distance remains resulting in an uncertainty when comparing extinction coefficients directly. An additional uncertainty arises from the fact that the LIDAR ratio (extinction coefficient/ backscatter coefficient) is not known. The in-situ data may help to constrain the LIDAR ratio, which will be tested in future with more joint data from our ongoing campaign."*

*To:*

*"Whenever LIDAR measurements … a Raman-Mie LIDAR (Blum 2005). When retrieving the extinction coefficients from LIDAR measurements, an uncertainty arises from the fact that the LIDAR ratio (extinction coefficient divided by backscatter coefficient) is not known. An additional uncertainty stems from the fact that the LIDAR beam and B-ICI do not sample the cloud at the same location. Even though air parcels sampled by the LIDAR may advect close to the balloon trajectory, a certain distance usually remains. Despite these uncertainties, a preliminary study found that the extinction coefficients retrieved from LIDAR measurements compare favourably with the extinction measurements of B-ICI (Kuhn 2017). However, here we have not retrieved the extinction coefficient, and the remaining distance between LIDAR and B-ICI measurements has not been considered. In this study only*

*the backscattered signal is used directly as complementary information to assess the temporal and spatial characteristics of the ice clouds sampled with B-ICI."*

**And one sentence has been moved to the end of Summary and Conclusions:**

*The in-situ data may help to constrain the LIDAR ratio, which will be tested in future with more joint LIDAR and B-ICI data from our ongoing campaign.*

- page 9, line 14: The defition of liquid origin vs. in situ ice clouds is still not complete (or even correct), especially the thermodynamic aspect (at approx. water saturation vs. below water saturation) is not mentioned. My suggestion would be: 1. in situ: homogeneous freezing of solution droplets or heterogeneous nucleation (deposition nucle- ation) for RH < 1 and T < 235 K 2. liquid origin: homogeneous/heterogeneous freezing of pre-existing cloud droplets for RH ~ 1 and T > 235 K. Please expand the text in this respect.

**Thank you for the clarification. We actually tried to keep the description of the classification as simple as possible. For the difference in the microphysical properties between liquid and in-situ origin cloud we found, the exact formation pathway is not that important. It is more the temperature origin and available water vapour. But to be more precise we have added the relative humidity wrt. to water and ice to the text, which now reads as follows:**

*"This classification is described in detail by Krämer et al. (2016) and Luebke et al. (2016) and is briefly outlined in the following. The classification is based on cloud origin temperature: 1) In-situ origin cloud: homogeneous freezing of solution droplets or heterogeneous (deposition nucleation) at temperatures < 235 K (with $RH_{water} < 1$, and $RH_{ice} > 1$); 2) liquid origin cloud: heterogeneous or homogeneous freezing of pre-existing cloud droplets at temperatures > 235 K or ~235 K ($RH_{water} < 1$ or ~1 and $RH_{ice} > 1$). In case of a liquid origin cloud, the ice particles formed at lower altitudes via the liquid phase and were lifted subsequently to the cirrus temperature range (< 235K)."*

- Figures 5,7,8: For a better representation, you could use 2 blocks of $2 \times 2$ images for each ice cloud category.

**Thank you for this advice. Figures 7 and 8 are now in 2x2 blocks, Figure 5**

**has been rotated 90 degrees to allow for larger plots.**

- page 15, line 33 and following text:I don't think this explanation is completely correct. Even for very low temperatures ice crystals can grow to larger sizes and complex shapes (see e.g. fig. 5, lower panel, in Bailey and Hallett, 2009), if they get enough water vapour. The main issue here is the competition for water vapour, i.e. if many ice crystals compete for the available water vapour, they cannot grow to large/complex sizes. This is probably the case for 2013-02-20, since the number concentration of ice crystals is huge. Please change the text accordingly.

**We agree that particles can grow large even at low temperatures. This can happen when only a few ice particles form heterogeneously and live long in slow updrafts, and then the water vapour is sufficient. Jensen et al. 2008 showed that at 186K and H2O ~ 3 ppmv ice crystals can still grow up to ~ 100 µm.**

**Changed from:**

*"In-situ origin clouds form at a temperature range (< -38°C) where the water concentration in the atmosphere is very low. Therefore, there is not enough water available to form very large or complex shapes. Hence, it is understandable that most of the in-situ origin cirrus particles found were compact."*

**To:**

*"In-situ origin clouds form at a temperature range (< -38°C) where the water concentration in the atmosphere is much lower than at the warmer temperatures at which ice forms in liquid origin clouds. This, combined with the usually higher number concentrations of ice particles formed in in-situ origin clouds (fast updrafts) and the resulting competition for water vapour, does not allow to form very large and complex particles. Hence, it is understandable that most of the in-situ origin cirrus particles found were compact. It should be noted that in some cases large ice crystals can form also in in-situ origin clouds under calm (low vertical updrafts) or clean conditions (low INP concentration). In these cases, the number concentration would be very low, which in turn allows the ice crystals to grow to larger sizes (e.g. Jensen 2008). However, this type of cirrus cloud is not dominating our observations or the large collection by Krämer et al. (2016) and Luebke et al. (2016)."*

- Technical comments Figure 9, left panel: wrong key, it should read "2016-02-12".

**Thank you, we have corrected that.**

[revised manuscript text omitted]
 divided by backscatter coefficient) is not known. An additional uncertainty stems from the fact that the LIDAR beam and B-ICI do not sample the cloud at the same location. Even though air parcels sampled by the LIDAR may advect close to the balloon trajectory, a certain distance usually remains. Despite these uncertainties, a preliminary study found that the extinction coefficients retrieved from LIDAR measurements compare favourably with the extinction measurements of B-ICI (Kuhn et al., 2017). However, here we have not retrieved the extinction coefficient, and the remaining distance between LIDAR and B-ICI measurements has not been considered. In this study only the backscattered signal is used directly as complementary information to assess the temporal and spatial characteristics of the ice clouds sampled with B-ICI.

[revised manuscript text omitted]

The classification is based on cloud origin temperature, i.e. the temperature at formation of the cloud:

1) In-situ origin cloud: homogeneous freezing of solution droplets or heterogeneous (deposition nucleation) at temperatures $< 235\,\mathrm{K}$ (with $RH_{water} < 1$, and $RH_{ice} > 1$);

2) liquid origin cloud: heterogeneous or homogeneous freezing of pre-existing cloud droplets at temperatures $> 235\,\mathrm{K}$ (with $RH_{water} < 1$ and $RH_{ice} > 1$). In case of a liquid origin cloud, the ice particles formed at lower altitudes via the liquid phase and were lifted subsequently to the cirrus temperature range ($< 235\,\mathrm{K}$).

As formation in this context, we consider the time when the ice water content (IWC) started to be greater than zero, or 24 h before the in-situ measurement in case IWC was greater than zero during these 24 h. Consequently, the cirrus origin is determined here using temperature and IWC along 24 h back trajectories. The Lagrangian microphysical model CLaMS-Ice (Luebke et al., 2016) is used to calculate these trajectories, starting from locations along the balloon flight paths, based on ECMWF ERA-Interim meteorological fields as input. Temperature is interpolated onto the trajectories, whereas the IWC along the trajectories is simulated with CLaMS-Ice. The origin of an observed cirrus cloud is identified as in-situ origin if the temperature of the trajectory is always below 235 K. In case the temperature was originally higher than 235 K and the trajectory carried already IWC at the time temperature crossed 235 K towards colder values, the observed cirrus is assigned as liquid origin. The resulting classifications are listed in Tab. 1. Half of the measured cirrus clouds are classified as in-situ origin, the other half as liquid origin.

**3.3 Cloud properties**

The cloud extent and averaged temperature for each measurement are listed in Tab. 1, and Fig. 5 shows the corresponding temperature and humidity profiles. Two cirrus clouds (2015-04-01 and 2016-02-12) have a vertical extension of approximately 6 km with a low cloud base at an altitude of 2 km and 3 km, respectively. It may not be correct to call these clouds cirrus. However, in both cases, the entire cloud contains ice phase only, and the lower level represents, as will be discussed later, a glaciated, previously mixed-phase cloud layer. We believe these to be interesting cases and included them in our cirrus study. The other six cirrus clouds are thinner (80 m – 2 km thick) and had a higher cloud base (over 6 km). In all cases the temperature decreases with altitude. The temperature at the cloud tops is between -60° C and -70° C. At the cloud bases, the temperature is between -45° C and -55° C in case of thin clouds and between -10° C and -20° C in case of the two thick clouds. The relative humidity with respect to ice in the clouds is between 80% and 130%. Particles with sizes between 10 µm and 1200 µm have been collected. Smaller particles are not efficiently sampled (Kuhn and Heymsfield, 2016), and larger particles have not been encountered. Table 2 lists the size ranges and mean number concentration for each cloud. The ice particle number concentrations are between (3 / L) and (400 / L) and the profiles of the number concentration for each measurement day are shown in Fig. 8. For each measured cirrus cloud, the frequency of occurrence of shapes is summarized in Tab. 2 as percentages corresponding to the five particle shape groups compact, irregular, rosettes, plates and columnar. Some images of the particles from each measurement are shown in Fig. 6. All particle images are shown with the same size scaling.

[Figure]

**Figure 4.** Extinction coefficient in km$^{-1}$ (left) derived from IRF LIDAR and vertical velocity in m/s (right) obtained from ESRAD on 20.2.2013.

**Table 2.** List of days, mean number concentration $n$, mean particle $D_{max}$, and relative number (in %) of particles in different shape groups.

| Date - origin | $n$ | $D_{max}$ min/ median/ max | compact | irregulars | rosettes | plates | columnar |
|---|---|---|---|---|---|---|---|
| | 1/L | µm | % | % | % | % | % |
| 2012-04-04 - in-situ | 37 | 7/ 96/ 327 | 38.8 | 42.2 | 16.4 | 0.4 | 2.2 |
| 2013-02-20 - in-situ | 228 | 14/ 34/ 91 | 71.7 | 19.3 | 2.5 | 1.8 | 4.7 |
| 2016-03-15 - in-situ | 13 | 25/ 41/ 105 | 72.4 | 21.3 | 4.6 | 0.7 | 1.0 |
| 2016-12-15 - in-situ | 3 | 25/ 52/ 102 | 63.9 | 16.8 | 4.2 | 0.0 | 15.1 |
| 2013-12-18 - liquid | 16 | 24/ 84/ 277 | 4.3 | 69.6 | 6.5 | 8.7 | 10.9 |
| 2014-03-20 - liquid | 38 | 11/ 100/ 492 | 27.2 | 60.8 | 9.5 | 0.3 | 2.2 |
| 2015-04-01 - liquid | 8 | 22/ 201/ 643 | 7.2 | 23.9 | 49.3 | 1.6 | 18.0 |
| 2016-02-12 - liquid | 6 | 5/ 244/ 1228 | 6.0 | 39.2 | 28.5 | 3.7 | 22.6 |

[Figure]

**Figure 5.** changed to landscape orientation, bigger axes labels Temperature (red) and relative humidity profiles (blue) with respect to ice for the eight measurement days (upper row in-situ origin, lower row liquid origin). The clouds upper and lower edge are marked by horizontal lines.

[Figure]

**Figure 6.** Some pictures of ice particles from all measurement days. The left panel shows ice particles from in-situ origin cirrus, on the right liquid origin crystals are displayed. For a better understanding of the size, a 100 μm bar is displayed (2012-04-04 bottom). All images have the same scale resolution and 100 μm corresponds to 61 pixel.

**4 Results and Discussion**

**4.1 Size and Number concentration**

Section 4.1 has been revised and repetitions removed

Our measurements show differences in particle size and number concentration depending on cloud origin. This difference in size is also reflected in the number size distribution (PSD). Figure 7 shows PSDs for all measurement cases, where possible also for different height levels. In cases of in-situ origin cirrus, all particles are smaller than 350 μm. On three of the four days with in-situ origin (2013-02-20, 2016-03-15 and 2016-12-15) all particles are even smaller than 100 μm. The in-situ origin PSDs are fairly narrow, which indicates that the corresponding clouds were quite young, homogeneously formed cirrus, where the ice crystals have not yet grown to larger sizes. On 2012-04-04, the ice particles may have grown somewhat more than on the other days with in-situ origin clouds, leading to somewhat wider PSDs on that day. All distributions of the liquid origin clouds extend to larger sizes and are broader than in the case of the in-situ origin. In order for the particles to grow that large, a sufficiently high temperature with the related high water vapour concentration is required. Such conditions are given for liquid origin clouds. The PSDs of the two liquid origin clouds, originating from the south (2015-04-01 and 2016-02-12) with low cloud base (totally frozen, previously mixed-phase cloud) and large vertical extension, are particularly wide. On these two days (see Tab. 2 and Fig. 7) we have collected very large ice particles, with maximum sizes of approximately 600 μm and 1200 μm respectively. The other two liquid origin clouds have almost similarly narrow PSDs as the in-situ origin clouds and have probably been already in the process of dissolving, and large particles lost via precipitation. On 2013-12-18 for example, the cloud is very thin (80 m) and the relative humidity (ice) above and below the cloud is even strongly under-saturated. Many of the collected particles look as if parts have already sublimated. Data reported earlier from aircraft measurement at high latitudes also show a large range in sizes comparable to the observations of our balloon measurements. Gayet et al. (2007) described a measurement in which they collected falling ice particles from a cirrus cloud above. The size distribution between 25 μm and 1000 μm mentioned by them corresponds well with our two thick liquid origin clouds, which had their lower edge approximately at the same height and similar temperature. Furthermore, Sourdeval et al. (2018) presented PSDs from five aircraft campaigns (tropics: ATTREX, ACRIDICON-CHUVA; mid-latitudes: SPARTICUS, ML-CIRRUS and COALESC). In the averaged data, they observed, in addition to a primary mode of sub 100 μm particles that was always present, a secondary mode of larger than 100 μm particles that appeared only at temperatures higher than -50°C. They discussed that this large particle mode was due to liquid origin cirrus. Thus, a comparison with our measurements shows that in the Arctic liquid origin clouds with larger particles can still occur at lower temperatures.

In general, the PSDs are more narrow and the number concentration ($n$) higher with increasing height and decreasing temperature. This can be clearly seen, for example, for the in-situ origin clouds in Fig. 7 and Fig. 8. The dependence of in-situ origin PSDs and $n$ on altitude and temperature is likely due to the main ice nucleation zone being at the cloud top. This dependence of the PSDs and $n$ has also been found on a global scale (e.g. Sourdeval et al., 2018; Gryspeerdt et al., 2018). This PSD and $n$ trend with altitude and temperature is not clearly seen for the liquid origin cirrus cases. This could be explained by the fact that liquid origin cirrus form at lower altitudes and then ascent in the prevailing updraft. The few ice particles,

nucleated in warmer and thus also moister air masses, grow to large sizes which sediment out of the air mass while ascending. This means that in pure liquid origin cirrus there is no process enhancing the number concentration of smaller ice particles towards the cloud top or higher altitude.

However, these variations in PSDs with altitude or temperature are less than the general differences observed between in-situ origin and liquid origin. The broadest size distribution at the lowest height of the in-situ origin cloud on 2013-02-20 for example is still much more narrow than any distribution of the liquid origin cloud on 2016-02-12. This is true also for the other measurement days, even when considering the two liquid origin clouds with more narrow PSDs, which are still broader than in-situ PSDs at similar temperatures. That means that size distributions measured in different clouds but at similar altitudes and temperatures can be significantly different. While these differences are obviously not only related to the local ambient conditions, they are strongly related to the cloud origin.

The total number concentrations are shown in Fig. 8 as altitude profiles for all eight measurement days (top: in-situ origin, bottom: liquid origin). The number concentrations of the liquid origin clouds are relatively low ($5/L$ to $70/L$). The lower $n$ in comparison to Luebke et al. (2016), who found a median ice number concentration slightly above $100/L$ in liquid origin mid-latitude cirrus, might be due to a lower number of ice nucleating particles (INP) in Arctic regions (Costa et al., 2017), which are necessary for heterogeneous freezing. However, low number concentrations could also be caused by a dissolving cloud state. To confirm this, one would need INP or humidity measurements during some time before our measurements, hence, we can only speculate here.

Comparing 2013-02-20 and 2016-02-12, it can be seen that the maximum concentration of the in-situ origin cloud is approximately 20 times greater than in the liquid origin cloud. This much higher number concentration in the case of this in-situ origin cloud does not apply to all of our in-situ origin cases, but only to two measurements (2013-02-20 and 2012-04-04). On 2016-03-15 the $n$ is similar as in case of liquid origin measurements and on 2016-12-15 the $n$ is very low with just $3/L$. Such high differences in $n$ between in-situ origin clouds may be related to the influence of wave activity. The two days (2013-02-20 and 2012-04-04) with higher number concentration ($300-400/L$) and also 2016-03-15 are connected to very strong winds coming from the north-west that have led to waves, which have been observed by ESRAD or LIDAR on both days. These waves with the related high vertical velocities can be the needed trigger for such high number concentrations (e.g. Lohmann and Kärcher, 2002). Field et al. (2001) showed that number concentrations in wave clouds can even rise with decreasing temperature up to $100000/L$.

In contrast, the mid-latitude in-situ origin cirrus observations, described by Krämer et al. (2016) and Luebke et al. (2016), showed lower $n$ than those in the Arctic. The reason may be that most of the Arctic in-situ observations are influenced by waves with high vertical velocities triggering homogeneous nucleation of many ice crystals. Such observations were very rare in Krämer et al. (2016) and Luebke et al. (2016).

Krämer et al. (2016) discussed two types of in-situ origin cirrus. The first type appears in slow updrafts, e.g. in warm conveyor belts. The ice is nucleated mostly heterogeneously and the corresponding ice particle number concentrations are in the range of the available INP and rather low. This applies to the majority of the measurements analysed by Krämer et al. (2016) and Luebke et al. (2016) and to one of our measurement days (2016-12-15).

In the second type which is related to fast updrafts, the ice particles formed by homogeneous nucleation. This is reflected in a high number concentration and is more often the case in our measurements. As a result, in our case three of the in-situ origin clouds have higher or about the same $n$ as liquid origin clouds.

Liquid origin clouds are present typically in case of convection or large scale transport like warm conveyor belts. The cirrus ice particles of liquid origin are mostly formed by heterogeneous freezing at lower altitudes and temperatures above 235 K, where typically mixed-phase clouds occur. They are uplifted into the in-situ temperature range where they at latest fully glaciate. In the original altitude, more water vapour and INPs are available, resulting together with the continuous updraft in larger particles, higher number concentration and thus higher ice water content compared to the in-situ origin clouds with slow updraft. As indicated earlier, differences between mid- and high latitudes may then be explained by differences in the available INP for liquid origin clouds and the larger influence of waves on our measurement cases in the Arctic in case of in-situ origin.

**4.2 Shape**

In our individual cases there is no significant dependence of the shape on temperature and relative humidity (with respect to ice). Furthermore, no particular dependence of particle shape over the height was found. Therefore, we are reporting the average frequency of occurrence of the different particle shapes (see Tab. 2) and discuss how that varies depending on cloud origin. These average frequencies of shape occurrence for in-situ origin and for liquid origin clouds are also shown in Fig. 9 (left panel). The right panel of this figure shows how the average particle sizes of the different shapes vary depending on the cloud origin. As can be seen in Fig. 6 and Fig. 9, in the case of in-situ origin, the particles are usually small in size and compact or irregular in shape. However, in the case of liquid origin, the particles are most commonly irregular and rosettes.

In-situ origin clouds form at a temperature range (< 235 K) where the water concentration in the atmosphere is much lower than at the warmer temperatures at which ice forms in liquid origin clouds. This, combined with the usually higher number concentrations of ice particles formed in in-situ origin clouds (fast updrafts) and the resulting competition for water vapour, does not allow to form very large and complex particles. Hence, it is understandable that most of the in-situ origin cirrus particles found were compact. It should be noted that in some cases large ice crystals can form also in in-situ origin clouds under calm (low vertical updrafts) or clean conditions (low INP concentration). In these cases, the number concentration would be very low, which in turn would allow the ice crystals to grow to larger sizes (e.g., Jensen et al., 2008). However, this type of cirrus cloud is not dominating our observations or the large collection by Krämer et al. (2016) and Luebke et al. (2016).

While compact particles are on average the smallest ones, rosettes, irregular and columnar particles in liquid origin clouds are largest. Liquid origin cirrus clouds, in contrast to in-situ origin cirrus, form at warmer temperatures with higher water vapour content in the air. Therefore, the ice particles can grow larger and also to more complex shapes. Particularly large ice particles have been observed on the two days (2015-04-01 and 2016-02-12) where the lower part of the cloud was in the temperature regime of mixed-phase clouds. As discussed earlier, at the time of measurement these two clouds were completely frozen. However, the liquid water, which was probably present at some earlier stage, has contributed to the observed extensive growth. This is in agreement with Bailey and Hallett (2009), who claimed that a high supersaturation is needed for the growth of rosettes and hollow columns, which are abundant on those days. Fewer rosettes are found on the other two days (2013-12-18

[Figure]

**Figure 7.** new order of panels  Number size distributions for all measurement days for different cloud levels (top: in-situ origin; bottom: liquid origin).

[Figure]

**Figure 8.** new order of panels  PSDs at different cloud levels for all measurement days (top: in-situ origin; bottom: liquid origin)

and 2014-03-20). This may be unexpected, however, it may be explained by larger particles falling out of the probably ageing clouds. In fact, these clouds look like they have been in the process of dissolving, as discussed earlier in Sect. 4.1.

It is noticeable that almost all columnar particles and rosettes are hollow in case of liquid origin cirrus. This corroborates findings by others (e.g. Weickmann et al., 1948; Heymsfield et al., 2002; Schmitt et al., 2006), in which measurements showed that around 80 % of all collected rosettes were hollow to a certain extent. In the case of in-situ origin cirrus there are very few, and if present, then very small rosettes and columns. Thus, a statement regarding their hollowness would be rather speculative.

The supersaturation present in our liquid origin cloud measurements is most of the time too low to directly explain growth of our observed hollow rosettes and columns. According to laboratory measurements by Bailey and Hallett (2004), existence of hollow rosettes requires high supersaturation, and hollowness of rosettes is more likely at higher temperatures (> -40°C). While the temperature and water vapour at which the particles have been detected is too low, ambient properties at the origin of the clouds met the conditions for hollow rosette growth in the case of liquid origin clouds. Thus, this demonstrates once more that environmental conditions at cloud origin are crucial for explaining observations.

In both origin cases, plates and columnar particles have been rarely collected. They are on average less frequent than any of the other shapes. This is similar to Korolev et al. (1999), who collected only 3 % of these shapes. In Tab. 2 it can be seen that in the case of in-situ origin clouds on 2013-02-20 the highest percentage of plates has been sampled with only 1.8 %. Somewhat more columns have been collected, on average 5.7 %. In the case of liquid origin these particle shapes are on average more frequent than in the case of in-situ origin, as can be seen in Fig. 9. Plates are on average 3.0 % of all observed ice particles, and columns are with 12.3 % even a little more frequent than compact particles (10.8 %).

Shape detection is sometimes intricate, even with high image resolution. Some particle shapes may be confusing, as also observed by others (e.g., Lindqvist et al., 2012). Here, the assignment between irregulars and rosettes is sometimes ambiguous, because in a few cases rosettes appear somewhat irregular. For example, some rosettes look as if they have a part missing or one bullet seems to be a longer column. In such cases, we have assigned these irregular rosettes to the shape group rosettes rather than to irregulars. In other cases, small compact ice particles sometimes show characteristics that indicate an initial formation of rosettes, however, we have still classified them as compact due to their spheroidal shape. Classifying them as rosettes would not have changed any of the results discussed here.

For ice particles smaller than 20 μm the shape is difficult to recognize and, consequently, some misclassification may occur leading to over-representation of compact in this size range and under-representation of other shapes such as plates and rosettes. However, on the day with the smallest particles (2013-02-20) only about 6 % of all particles are smaller than 20 μm. Thus, this issue of potential misclassification will likely not alter our findings significantly.

**5 Summary and Conclusions**

In this study, eight balloon-borne in-situ measurements of Arctic cirrus clouds were analysed. The balloons were launched from Kiruna, Sweden during winter time. Particular emphasis was placed on the analysis of ice particle size, shape and number concentration with respect to cirrus origin. Since in-situ origin clouds are formed from the gas phase at temperatures below

[Figure]

**Figure 9.** Occurrence of different particle shapes depending on cloud origin (left) and average $D_{\max}$ for the different shapes (right). Mean values of shape and size of the four in-situ origin (gray) measurements and four liquid origin (light blue) measurements. Mean values of shape and size of one in-situ origin measurement day (black) and of one liquid origin measurement day (blue).

235 K, while liquid origin clouds formed via liquid drops at temperatures above 235 K, the cloud and particle properties are expected to vary in accordance to cloud origin. Indeed, while large differences in particle size, shape and number concentration are observed between the various measurements, some similarities are noticed within the two groups of data with liquid and in-situ origin clouds, respectively. These similarities and the differences between data, when grouped in liquid and in-situ origin, are summarized below:

1. Particle size: Arctic cirrus clouds with particle sizes between 10 μm and 1200 μm have been observed. Most common in our clouds are particles with sizes between 30 μm and 250 μm. While in-situ origin clouds have smaller particles with sizes below 350 μm, liquid origin clouds exhibit larger particles and wider number size distributions. The ice particles of clouds with wind from the south are much larger and fewer than ice particles from the west where the cirrus was probably triggered by strong updrafts associated with waves behind the Scandinavian Mountains.

2. Particle shape: The in-situ origin clouds consist mainly of compact and irregular particles and the liquid origin clouds of irregular, rosettes and columns. In both cases, there are hardly any plates. The compact particles are the smallest particles and rosettes are the largest. Rosettes and columns are mostly hollow.

3. Particle number: The measured number concentrations are between 3/L and 400/L. Both extreme values have been determined for in-situ origin clouds. The maximum concentrations have occurred due to waves on the lee side of the Scandinavian Mountains. In comparison, in previous campaigns in the mid-latitudes lower number concentrations were measured for this cloud type. This may be explained by the fact that hardly any wave-induced in-situ origin clouds were observed in these campaigns. Concentrations for liquid origin clouds are low (5/L to 70/L). In contrast, high number concentrations were measured in the mid-latitudes for this cloud type, maybe caused by a higher number of INPs in the mid-latitudes than in the Arctic.

The results of this study imply that remote sensing retrievals and weather and climate models could be improved when accounting for these differences rather than using parameterisations that depend only on local conditions. Future work will include more measurements for further significant statistical evaluation. In addition, we also want to allow several B-ICIs to fly one after the other in order to investigate a temporal development of the particle properties. Also, the in-situ data may help to

5     constrain the LIDAR ratio, which will be tested in future with more joint LIDAR and B-ICI data from our ongoing campaign.

*Acknowledgements.* We thank Peter Dalin (IRF) and Evgenia Belova (IRF) for interpreting and discussing the ESRAD data in terms of gravity and mountain lee-waves. We thank the Swedish National Space Agency for funding these balloon campaigns (Grants Dnr 85/10, Dnr 86/11, Dnr 143/12, Dnr 273/12, Dnr 168/13, Dnr 124/14).

**References**

[revised manuscript text omitted]